# Scientific Theory of a Black-Box: A Life Cycle-Scale XAI Framework Based on Constructive Empiricism

## Abstract

Explainable AI (XAI) offers a growing number of algorithms that aim to answer specific questions about black-box models. What is missing is a principled way to consolidate explanatory information about a fixed black-box model into a persistent, auditable artefact that accompanies the black-box throughout its life cycle. **change:** In this conceptual work we address this gap by introducing the notion of a *scientific theory of a black box* (SToBB). Grounded in Constructive Empiricism, a SToBB fulfils three obligations: (i) *empirical adequacy* with respect to all available observations of black-box behaviour, (ii) *adaptability* via explicit update commitments that restore adequacy when new observations arrive, and (iii) *auditability* through transparent documentation of assumptions, construction choices, and update behaviour. We operationalise these obligations as a general framework that specifies an extensible observation base, a traceable hypothesis class, algorithmic components for construction and revision, and documentation sufficient for third-party assessment. Explanations for concrete stakeholder needs are then obtained by querying the maintained record through interfaces, rather than by producing isolated method outputs. **change:** To illustrate the framework we build a step-by-step example for a neural-network on a tabular task. Together, these contributions position SToBBs as a life cycle-scale, inspectable point of reference that supports consistent, reusable analyses and systematic external scrutiny.

## 1 Introduction

Explainable AI (XAI) provides methods for analysing black-box decisions. Each method builds on its own assumptions and evaluation criteria and is intended to answer specific questions for particular users (Speith, 2022; Tomsett et al., 2018; Nauta et al., 2023; Mersha et al., 2024). While XAI offers methods tailored to individual needs, these methods are typically applied in an ad hoc manner, producing isolated outputs that may need to be recomputed for similar questions and that can yield incompatible characterisations (Krishna et al., 2024) of the same model. To address this problem, we propose a perspective that treats explanatory information as an evolving artefact maintained across the black-box's *life cycle*, rather than as a series of disconnected method outputs regarded as independent of one another. From this view, such an artefact should ensure adequacy, adaptability, and auditability over time, providing a stable basis for consistent analyses while remaining open to new forms of inquiry. We argue that one way forward is to treat explainability for machine learning systems as playing a role similar to that of *scientific theories* in science: not as something designed to answer any particular question outright, but as a persistent and auditable consolidation of knowledge that provides a shared point of reference and supports diverse forms of inquiry. To this end, we propose and develop the idea of a *scientific theory of a black-box* (SToBB).

To make this idea more concrete, consider a fixed classifier trained on the Abalone dataset (Nash et al., 1994) that predicts whether an individual abalone has reached a target age, relevant to harvesting decisions and pricing. During development, model developers probe the classifier globally to understand which input features (e.g., physical measurements of shell size or weight) influence predictions and where the model fails, using global summaries and debugging-oriented explanations. After deployment, operators and employees in an aquaculture setting may request local or contrastive explanations for specific batches, e.g., why a given lot is classified as not ready for harvest and what changes in measurements would flip that assessment. At the

same time, sustainability officers or external auditors may periodically inspect the model's behaviour over time, asking whether different abalone subpopulations are treated consistently and whether model decisions reflect sustainability obligations. Wholesalers seeking to buy abalone may request justification for the asked price, which could be supported by explanations that demonstrate the relevant qualities of the batch, such as predicted age, size, and compliance with sustainability constraints. All of these inquiries concern the same underlying classifier and lead to similar probings of it, but arise in different contexts, i.e., different stages of the system's life cycle and from different stakeholders who vary in intent and background knowledge. To address a diverse set of inquiries, a SToBB organises shared explanatory information as a maintained record that can accumulate over time and be re-used in different contexts; concrete, context-appropriate explanations are then obtained by querying this record.

**change:** This paper argues that *Constructive Empiricism* (CE) (van Fraassen, 1980) and its conception of *scientific theories* offers a useful framework for XAI in application settings where explanatory information is treated as a maintained artefact that stays accountable to observed model behaviour over time, rather than as a one-off effort to recover the internal workings of a black-box model. ~~In particular, CE aligns with two central requirements in XAI: consistency with observed model behaviour and pragmatic usefulness to different users. In CE, scientific theories are judged by their *empirical adequacy,* that is, their agreement with all observed phenomena, and by their practical usefulness to those who employ them.~~ According to CE, acceptance of a theory rests on two commitments: belief in its empirical adequacy and a willingness to use and adapt it in new situations and as new observations arise. A theory is therefore part of a continuous process in which observations either reaffirm its adequacy or demand refinement. Notably, CE does *not* assume that a theory must correctly describe an underlying truth that lies beyond what is observable. **change:** In our setting, this maps directly onto the requirements above: empirical adequacy yields the adequacy obligation, the commitment to ongoing revision yields adaptability, and making both commitments transparent and assessable for third parties yields auditability. Faithfulness, in the sense of providing true descriptions of underlying mechanisms or reasoning processes, is not a requirement.

CQYP

CQYP

To operationalise the concept of a SToBB for XAI, *surrogate models* offer a natural starting point. Surrogates are inherently interpretable models designed to approximate the black-box's input-output mapping (Bastani et al., 2017; Molnar, 2020). They are typically applied post hoc and independently of the black-box's architecture (Speith, 2022) and may approximate behaviour locally (Ribeiro et al., 2016; Lundberg & Lee, 2017; Ribeiro et al., 2018) or globally (Bastani et al., 2017). ~~In contrast to scientific theories as understood in CE, surrogates are not generally required to achieve perfect agreement with the black-box on all observed data, nor are they systematically revised when inconsistencies emerge.~~ **change:** CE provides a principled basis for recasting surrogates as scientific theories: while a surrogate model does not pretend to capture any hidden, unobservable truth about what underlies the behavior of the black-box, it must agree with the black-box on all available observations as they accumulate if it is to support explanations, and when a new observation reveals an inconsistency, the surrogate must be revised to restore adequacy. This is stricter than typical surrogate practice, where error on observed data is tolerated and inconsistencies are not systematically addressed, and it is what distinguishes the maintained SToBB from a merely approximate aid.

CQYP

Scientific theories are also documented, scrutinised, and evaluated within a community. The same principle motivates a SToBB, which must make explicit its assumptions, construction choices, and update behaviour to enable third parties (e.g., auditors or operators) to independently assess its merit, scope, and limitations. Detailed transparency requirements and their connection to existing documentation practices are discussed in Section 2.

It is important to emphasise that we consider the SToBB a complementary perspective in XAI and not a universal solution. It addresses a specific gap: the lack of a persistent, expanding, auditable representation of explanatory information that accompanies a black-box as observations accumulate. As a result, SToBBs serve as a life cycle-scale, inspectable point of reference for explanation: empirical adequacy provides a checkable baseline, while decoupling explanatory information from its context-dependent presentation enables consistent, reusable analyses across stakeholders. Throughout this paper, we treat the black-box model as fixed, meaning that its decision behaviour does not change while explanatory information is collected. We return to this assumption and its implications in the discussion.

In this paper we make the following contributions:

- We introduce the notion of a SToBB, a documented and evolving artefact that represents a fixed black-box model via an interpretable surrogate, an observation base, and explicit update and documentation commitments.

- As a conceptual prerequisite to this definition, we operationalise CE for XAI by translating empirical adequacy, acceptance as commitment, and pragmatic virtues into concrete design and documentation obligations for constructing and maintaining SToBBs.

- **change:** As an illustration of the framework, we present a step-by-step example that instantiates a SToBB for a neural-network classifier on a tabular task.~~; to this end, we introduce the *Constructive Box Theoriser* (CoBoT) algorithm as a generic procedure for tabular domains, which may be of independent interest.~~

Outline: Section 2 motivates the structural gap in the current XAI landscape that a SToBB aims to address. Subsequently, Section 3 introduces the central notions of Constructive Empiricism and transfers the concepts to XAI. This yields an overview of the properties a SToBB must fulfill. Based on this, Section 4 defines the constituents of the SToBB and discusses its properties. Section 5 goes on to describe and exemplify the process to create a SToBB. Section 6 provides the discussion and conclusion.

## 2  Context

In this paper we argue for a perspective that treats explanatory information regarding a black-box as something that is collected and integrated over time. In this section we examine practical motivations for this perspective, focusing on two aspects: (i) heterogeneous stakeholder needs, and (ii) transparency needs that arise throughout the AI system life cycle.

**(i) Stakeholders and explanatory needs**  The XAI literature recognizes a diverse set of stakeholders who require insight into the behaviour of AI systems. These groups include, but are not limited to, developers, regulatory entities, operators, domain experts, and affected users (Tomsett et al., 2018; Barredo Arrieta et al., 2020; Langer et al., 2021; Mersha et al., 2024). Langer et al. (2021) emphasize that individuals may belong to several stakeholder groups and that individuals within a group can differ in expertise, background knowledge, and expectations. Research on user needs further shows that these stakeholders ask different kinds of explanatory questions and rely on different levels of abstraction (Liao et al., 2020; 2021).

**(ii) Transparency needs across the AI life cycle**  Stakeholder groups largely identical to those identified in the XAI literature also appear in several Trustworthy ML frameworks and standards.[1] All frameworks embed AI systems within a broader *life cycle* that spans design, development, testing, verification, deployment, and monitoring. Although the frameworks differ in focus – from risk management to organisational governance – they identify similar qualities for trustworthiness, such as accountability, human agency, technical robustness, compliance, and transparency. Importantly, transparency is regarded not only as a property in itself but also as an enabler for these other qualities, and transparency needs arise *continuously* at all stages of the life cycle (NIST, 2023; EU HLEG AI, 2019; OECD, 2024). Explainability plays an important role in addressing some of these transparency needs. As the Abalone example in the introduction motivates: Developers may rely on debugging-oriented analyses early in the life cycle, end-users may require local or contrastive decision explanations to verify fair treatment or obtain actionable information, and operators or auditors may need system-level evidence over time to assess compliance and orderly operation.

---

[1]IEEE P7001 (IEEE, 2022), ISO/IEC 42001 (ISO/IEC, 2023), the OECD AI Recommendation (OECD, 2024), the NIST AI Risk Management Framework (NIST, 2023), the Fraunhofer Guidelines for Trustworthy AI (Poretschkin et al., 2023), and the EU Ethics Guidelines for Trustworthy AI (EU HLEG AI, 2019) that now underpin the EU AI Act (European Parliament and European Council, 2024)

**The structural gap in the XAI landscape** While XAI acknowledges diverse stakeholders, the life cycle perspective is largely absent. Surveys of XAI methods (Barredo Arrieta et al., 2020; Speith, 2022; Mersha et al., 2024) show that research predominantly develops techniques for isolated explanatory tasks. Although these methods see successful practical use (Liao et al., 2021; Arya et al., 2019), they remain compartmentalised even when probing the *same* black-box model. In particular, explanations are produced independently of each other, which may lead to incompatible descriptions of model behaviour, a phenomenon known as the *disagreement problem* (Krishna et al., 2024), or to redundant computations over time.

Recall that stakeholders may have different explanatory goals and differ in their background knowledge. Despite these differences, all inquiries concern the same underlying model. If explanatory information is collected only for the immediate question at hand, it remains siloed and must be recomputed when similar questions arise later in the life cycle. A structured approach that records and consolidates explanatory information therefore enhances efficiency, consistency, and reuse across stakeholders. We propose to treat explanatory information not as a set of disconnected method outputs, but as a persistent *record* that aggregates observations and remains consistent as it evolves. Such a record is not itself an explanation, but a representational foundation from which diverse explanations can be derived. To the best of our knowledge, there is no artefact in the current XAI landscape intended to simultaneously (i) record, (ii) represent, and (iii) continuously update explanatory information for a fixed black-box as observations accumulate, while remaining auditable and reusable across the system life cycle. The SToBB framework fills this gap by treating explanatory information as a maintained record rather than a sequence of disconnected method outputs.

**Constructive Empiricism for XAI** What would such an artefact be? From the perspective of the philosophy of science, something that represents the behaviour of a system and serves as a common foundation from which diverse explanations can be derived is a *scientific theory*. ~~Following this perspective we turn, in particular, toward the operationalisation of CE CITEXX as a framework for structuring such a record. The semantic conception of theories in CE and the emphasis on empirical adequacy and updateability align closely with the requirements identified above. The next section introduces the core notions of CE and discusses their translation to XAI.~~ **added:** But scientific theories can be understood in different ways, and the choice matters. A realist framework would require the artefact to approximate the model's true internal reasoning, an unverifiable demand for any system whose internal workings are inaccessible by nature. The problem at hand calls for something more tractable: a framework whose epistemic commitments are grounded in what can actually be observed, checked, and measured. Constructive Empiricism (CE) (van Fraassen, 1980) is built around precisely this. Like the classical falsificationism of Karl Popper (Popper, 1959), CE holds that a scientific theory can only be tested against evidence from observation, and that a theory that makes false observational predictions should not be accepted. A theory earns acceptance only when it remains consistent with all available observations and makes explicit the conditions under which it would need to be revised. For a governance-oriented artefact that must be scrutinised by regulators and auditors, this directness is consequential: claims grounded in documented, observable behaviour can be checked and contested, whereas claims about unobservable internal reasoning cannot. However, whereas falsificationism is a purely epistemological view about the relation of observational evidence to scientific theory, CE goes beyond this by characterizing the significance of the theory for the scientist – which lies not in an attempt to reveal unobservable truths but in the theory's ability to provide a useful basis for prediction, exploration, and intervention. This fits well with the need we have identified for an artefact that can support various explanatory purposes over the life-cycle of the model. CE is compatible with competing views of how scientific theories themselves are revised or replaced over time, such as those of Thomas Kuhn (Kuhn, 1962; 1977) and Imre Lakatos (Lakatos, 1970; 1978). CE is neutral about the dynamics of theory-change. The emphasis in CE is on empirical adequacy, ongoing revision, and utility for explanatory needs. This aligns directly with the requirements identified above, and the next section develops this alignment in detail.

CQYP

**Documentation as a second strand of transparency** While some transparency needs are addressed by XAI methods, others are met by structured documentation aimed at stakeholders not directly involved in system development, e.g., to help understand a system's overall design, limitations, and intended use. Documentation frameworks exist that aim at a life cycle-scale solution (Chmielinski et al., 2024) or that support transparency for specific components of the ML pipeline: datasheets document datasets (Gebru

et al., 2021), model cards document trained models (Mitchell et al., 2019), and care labels summarise resource usage (Fischer et al., 2023). On the one hand, a SToBB itself serves as a piece of evolving documentation, falling in line with the works just mentioned but with a different focus. On the other hand, as the SToBB aims to remain useful throughout the black-box's life cycle, it itself will require accompanying documentation.

# 3   Adopting Constructive Empiricism to Explainable AI

In this section, we translate the central notions of CE into requirements for explanatory artefacts in XAI. This translation of requirements will serve as the conceptual basis for the *Scientific Theory of a black-box* (SToBB), introduced in Section 4. CE introduced by van Fraassen, is a philosophy of science that provides a definition of a scientific theory and a description of its relation to the world and to the scientist using it. A key feature of CE is that it does not require a theory to be literally true about the world, because the theory may include highly abstract elements or unobservable properties and processes whose correspondence with reality we cannot ascertain. Rather, a theory must be *empirically adequate* and pragmatically useful for scientific practice.

We will discuss *observational* and *relational substructures* as the two necessary constituents of a theory, and the notions of *empirical adequacy*, *theory acceptance*, and *pragmatic virtues*. For the reader's convenience, Table 1 aligns the central concepts of CE with their counterparts in our setting and already anticipates the design obligations that are defined formally in Section 4 and instantiated as a process in Section 5. It can already serve as a high-level guide for first-time readers, indicating the direction of subsequent sections, and it may also be referenced as a compact summary once the details have been introduced.

## 3.1   Observational and Relational Substructures

In CE's semantic view, a theory is presented as a family of structures. It distinguishes *observational* from *relational* substructures. Observational substructures represent the measurable states and outcomes accessible through instruments. Measurements may be erroneous and such observations may be rejected. *Relational substructures* in turn capture the rules or relations connecting those observations (think of variables and their composition to a formula). Together, they provide both the descriptive surface of a theory and the framework that supports explanation.

*Observational substructures:* For a SToBB, this requires making explicit what counts as an observation: the input variables, the target outputs, and the procedures by which they are measured. As we will later discuss, an observation must include, but need not be limited to, the inputs used by the black-box (Section 3.4).

*Relational substructures:* The semantic view of CE aligns with the notion of *surrogate models*: a hypothesis class forms a family of admissible structures, and each instantiated surrogate corresponds to a concrete structure within that family. For a SToBB, this requires specifying the surrogate's hypothesis class and the constraints imposed on it so that the mapping from inputs to outputs can be traced along an interpretable structure. We use *traceability* to refer to the ability to follow how observations are represented and processed by the surrogate in an interpretable way and to link specific surrogate components back to the observations that constrain them.

## 3.2   Empirical adequacy

According to CE, the central aim of a scientific theory is to be *empirically adequate*, that is, to be consistent with *all available* observations. A theory is empirically adequate if all observable phenomena can be represented within one of its observational substructures. This requires the theoretical ability to represent any possible measurement outcome as well as achieving internal consistency with all actually observed data. Importantly, CE does not claim that theories describe unobservable truths about the world; it is sufficient that they consistently account for all observed behaviour. No theory that conflicts with accepted observations can be advocated as correct. But there is no claim that the relational substructures represent the true reasoning that is being executed by the black-box or the hidden rules that really govern its behaviour.

| CE concept | CE definition | Constructing a Scientific Theory of a Black-Box |
|---|---|---|
| Observational substructures | The measurable states and outcomes accessible to science. | Define the observation space, including inputs, targets, possible auxiliary measurements derived from the black box and respective extraction methods. Make explicit what counts as an observation and how its quality is ensured. |
| Relational substructures | The rules or relations acting on observations. | Specify the hypothesis class of the surrogate and the structural constraints that make its input–output mapping interpretable. |
| Empirical adequacy | All observable phenomena must be representable and consistent with the theory. | Ensure that the surrogate reproduces the black box on all available observations. Adequacy is binary and must be restored through updates whenever inconsistencies appear. |
| Theory acceptance | Belief in empirical adequacy and commitment to its continued use. | Ground belief in demonstrable agreement with all observations and define a clear update policy to preserve adequacy when new data arise. |
| Pragmatic virtues | Criteria to choose between empirically adequate theories on grounds of usefulness. | Identify and justify pragmatic criteria shaping the surrogate, including user-centered design choices (e.g., compactness, coherence, comprehensibility), auxiliary observables added for pragmatic reasons, and documentation with diagnostic measures that record adequacy, adaptation, and design evolution. |

**Table 1:** Conceptual alignment between CE and the design obligations of a SToBB. It summarizes how each CE concept informs both structure and process.

Transferring this to surrogate models in XAI, the surrogate must reproduce the black-box on all currently available observations, but it needs not give guarantees for unobserved black-box behaviour. This is stricter than typical surrogate modelling in XAI, where approximation error on observed data is tolerated, and looser than standard ML practice, where performance on a held-out test set is used as a proxy for generalisation to an unknown data distribution. In the context of a SToBB, the black-box represents a fixed target function, and the task is not to generalise beyond it but to reproduce its behaviour on all available observations.

**added:** The stricter requirement is deliberate. For a SToBB, the surrogate is not one explanatory tool among many but the maintained descriptive reference from which explanations for the black-box are derived. In that role, a known mismatch means the artefact already documents its own failure to describe observed behaviour – it is not merely imperfect but already known to be wrong. Empirical adequacy therefore functions as a sharp maintenance criterion: any observation that falsifies the surrogate obliges an immediate revision.

CQYP, aMTd

**added:** When generalisation is not yet possible, the surrogate falls back to singleton representations – covering exactly one observation. This is the correct outcome, not a failure mode. A singleton makes explicit that an input-output pair has been observed but that no broader pattern is yet warranted. A non-adequate surrogate that places the same observation in a generalising rule it is known to violate appears more informative, but is already documented to be wrong. For a lifecycle-scale explanatory record, an adequate singleton is strictly preferable: it is less informative but never knowingly wrong, and it makes the limit of current knowledge explicit rather than concealing it.

CQYP, aMTd

### 3.3 Theory acceptance

For a scientist to *accept* a theory means (a) to believe that it is empirically adequate, and (b) to commit to using it. Committing entails a willingness to work with the theory in the expectation that it will be applicable to future observations and in new explanatory contexts, and to adapt it if inconsistencies arise. In van Fraassen's view, theory building is an iterative process informed by observations and testing.

While empirical adequacy is concerned with the relation between the theory and the world, acceptance pertains to the relation between the theory and the scientist. We see the acceptance of a theory in CE as analogous to accepting a model or an explanation. Accordingly, a SToBB needs to give grounds to justify its acceptance, i.e., the belief in its empirical adequacy and the decision to commit to its use. Belief in adequacy is grounded in the surrogate *demonstrably agreeing* with all available observations; the traceability requirement of the hypothesis class ensures that this agreement can be checked by linking surrogate behaviour back to specific observations. The commitment to use the theory extends this stance into the future by requiring efforts to maintain the surrogate's adequacy: if new observations reveal inconsistencies, the surrogate must be adapted to restore adequacy and the adaptation policy must be outlined comprehensibly. Thus, under CE, grounds for trusting the surrogate shift from one-off predictive performance on unseen data to transparent and continuous process-based guarantees.

**added:** The commitment to use the theory also points toward active testing. Singletons and under-populated regions of the surrogate, as discussed in the previous subsection, indicate where the empirical base is thin and where targeted observations would be most informative. A practitioner may deliberately probe such regions: an observation that confirms the current surrogate broadens the empirical foundation and strengthens the basis for trust; one that falsifies it triggers a documented update. Either outcome advances the epistemic state of the SToBB. 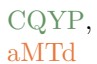

CQYP,
aMTd

### 3.4 Pragmatic virtues

Since multiple empirically adequate theories can coexist, CE recognises additional criteria for preference. If faced with two empirically adequate theories, it is rational to prefer one over the other solely on the basis of *usefulness*, whatever this may entail for the scientist. Properties that are determined to be relevant to this end are subsumed under the notion of *pragmatic virtues.* This may for example include choice of language, simplicity constraints, or other features that aid scientific inquiry. Pragmatic virtues do not concern the truth of the theory but only its practical value to the scientist.

In the context of a SToBB, two broader categories of virtues are particularly relevant:

(i) *User-centred criteria.* With the goal of being useful for explanation, stakeholder-centred criteria enter already at the design stage, where several choices must be made. For example, a general consensus exists regarding which broader hypothesis classes are considered inherently interpretable (Molnar, 2020; Barredo Arrieta et al., 2020). Designers may take these as inspiration when specifying the surrogate of a SToBB. Each hypothesis class is typically associated with established explanation formats for which further characterisations exist that describe desirable properties (Nauta et al., 2023). With the goal of later supporting explanation, the designer might therefore consider how their chosen hypothesis class could best enable such desirable properties.

Another important aspect is the inclusion of *auxiliary measures* in the observation space that quantify black-box behaviours beyond the raw prediction. The motivation for this arises from the Rashomon effect: multiple surrogates can achieve perfect fidelity to the black-box while relying on different internal logics (Breiman, 2001; Marx et al., 2020). Extending the observation space with auxiliary measures, such as feature-attribution information or other observations derivable from the black-box, constrains adequacy on a richer set of observables, reducing this multiplicity. Such enrichment does not reveal whether the surrogate's reasoning coincides with the "real reasoning" that underlies the behaviour of the black-box, but it adds observable intermediates that make the surrogate's behaviour empirically testable at finer resolution. From a CE perspective, this is a pragmatic virtue: it increases empirical discipline and user confidence without claiming access to unobservable reasoning.

(ii) *Documentation.*  A SToBB is intended to serve not only its creators but also external stakeholders such as operators, auditors, domain experts, and affected users. To enable independent judgment and informed use, a SToBB should therefore be accompanied by documentation, similar to model cards and datasheets mentioned in Section 2. It has to provide a detailed account of information related to its design, as to make assumptions, intended use, and scope explicit. It should detail how adequacy of the surrogate is determined, how its update process operates, and how other pragmatic design choices affect its behaviour. Analogous to how standard ML practices report learning curves, model size, or stability metrics to characterise training dynamics, the documentation can include diagnostic measures of its surrogate that record the evolution over time, e.g., how often new data triggers updates or how its structural components take shape. These diagnostics are not part of empirical adequacy but complement the descriptive documentation with quantitative evidence.

Utilising this alignment of CE to XAI concepts, we introduce SToBB, a novel artefact, in the next section.

# 4 Constituents and Properties of a SToBB

The previous section derived a set of concrete obligations from Constructive Empiricism (CE) and explained how they translate into operational requirements for XAI components. We now bring these elements together to define a SToBB and describe the properties that follow from its structure. Section 5 will discuss and exemplify the construction of such an artefact.

**Constituents.** A *scientific theory of a black-box* (SToBB) is a structured artefact that represents the behaviour of a fixed black-box through an interpretable surrogate and the processes that maintain it. A SToBB consists of the following five components:

1. **Observation base:** An extensible record of black-box input-output behaviour and auxiliary measures. These observations form the empirical basis that the surrogate must match to demonstrate empirical adequacy.

2. **Hypothesis class:** The specification of a traceable model family that operates on the observation space.

3. **Algorithmic components:** Procedures for *constructing* an initial surrogate and for *updating* it when new observations are added, ensuring that empirical adequacy is established and maintained over time.

4. **Adequate surrogate:** A concrete model instantiated from the hypothesis class that agrees with all records in the observation base.

5. **Documentation:** A description of all theoretical, functional, and operational properties and requirements, sufficient to enable a third party to audit, deploy, and fully maintain the SToBB.

Each component of a SToBB is required to satisfy one or more obligations derived in Section 3. In brief: *Empirical adequacy* requires an explicit observation base, an instantiated surrogate, and a clearly defined adequacy criterion. *Acceptance as commitment* requires an update procedure that restores adequacy when new observations arrive, and a hypothesis class that constrains the form of revisions. *Pragmatic virtues* require interpretable structure, diagnostics, and documentation that support transparency, auditability, and structured use.

## 4.1 Interfaces: Extracting explanations

We deliberately define the role of a SToBB to be the collection and representation of explanatory information in an empirically adequate and accessible manner; it is *not* its task to answer any particular question outright. When a concrete explanatory need arises for a given stakeholder and context, the information in the SToBB is accessed through *interfaces*. An interface is a procedure that takes the current SToBB and a query and

returns an answer tailored to that concrete application demand. It is analogous to a way of accessing a scientific theory or applying it for some specific purpose. Thus, interfaces are not core components of a SToBB, and the SToBB does not need to anticipate all future interfaces. They are important to the use of a SToBB but not for the conceptualisation in this paper. The minimal requirement for the SToBB is that the documentation is detailed enough to enable third parties to implement interfaces for their own use-cases.

As to not undermine the intent of the SToBB, a few general constraints for interfaces seem justified: To provide context-appropriate answers, interfaces may transform and structure information contained in the SToBB and add background information that the explainee requires. Since interfaces may answer questions regarding previously seen as well as unseen samples, newly encountered samples can be inconsistent with the current SToBB. In such cases, interfaces must not directly modify the SToBB; instead, an update must proceed via the documented update procedure. **added:** This means that querying the SToBB on unseen inputs is also a form of active testing: the result either confirms the surrogate on new evidence or triggers a traceable revision. To preempt the disagreement problem, an interface should, if possible, preserve in its output the traceability to both the underlying observations and the surrogate structure.

In line with pragmatic virtues, interfaces defined over time may be included with the SToBB. In that case, they should themselves be described in the documentation (purpose, assumptions, and dependencies on the SToBB), so that their purpose and behaviour are transparent for third parties. Creators of a SToBB will likely design interfaces for their own use, which may serve as useful examples for later maintainers.

## 4.2  Properties of a SToBB

The following properties stem directly from the definition of a SToBB, making it particularly valuable for XAI practice by providing a maintained and auditable point of reference across the black-box's life cycle.

*Baseline quality guarantees.* Because the surrogate is required to be empirically adequate, any analysis derived from it is grounded in a model that reproduces all known behaviour of the black-box. Interfaces may impose additional quality checks, but these build on a demonstrably consistent foundation.

*Extensibility.* The usefulness of the SToBB is not limited to only the use-cases its designers anticipated. Explanations are obtained through interfaces defined over the surrogate and its documentation. New types of analyses can therefore be introduced without rebuilding the SToBB, as long as they rely on the existing structures.

*Accumulation of knowledge.* As observations accumulate, the SToBB is strengthened because adequacy must hold for an increasingly rich empirical base. Existing results remain available, and new analyses can reuse previously established structure, which is especially valuable in time-sensitive settings.

*Traceability.* Every structural element of the surrogate can be linked back to the observations and update steps that shaped it. Explanations produced through transparent interfaces can likewise be traced to specific components of the SToBB, forming a continuous chain of reference to underlying data.

*Common point of reference.* Because all interfaces draw from the same surrogate and documentation, their results are comparable and interpretable in a shared context. If discrepancies arise between interface outputs, the transparency of the SToBB provides a principled basis for diagnosing and resolving them.

These properties collectively illustrate how a SToBB offers a coherent foundation for explanation with an emphasis on usefulness throughout the life cycle of the black-box. They integrate the obligations of CE: empirical adequacy as the minimal requirement, acceptance through transparency and a commitment to continued adaptation, and pragmatic virtues as criteria that guide design. The next section describes and exemplifies the practical process by which a SToBB is constructed and maintained.

## ~~5 Process and Proof of Concept: Constructing a SToBB via Constructive Box Theoriser~~

## 5   change: Constructing a SToBB: Process example

Section 4 defined a SToBB through five components. We now describe a process by which a researcher may construct this artefact in practice and how it is maintained over the life cycle of a black-box. To make the process concrete, we exemplify each component immediately in a proof-of-concept for a small neural network trained on a tabular task. ~~We introduce the *Constructive Box Theoriser* (CoBoT) algorithm and use it as the algorithmic component to build a SToBB. Note that the formal description of CoBoT is in SECREFXX, this section provides a high-level description as to not distract from the overall proof-of-concept.~~

The subsections are organised around the SToBB components *observation base*, *hypothesis class*, *algorithmic components*, and the resulting *adequate surrogate*. In each subsection, we first state what must be specified or created, and then illustrate how this is realised in the running example. Documentation aspects are discussed within the respective component subsections. In addition, we discuss operational requirements, as well as the optional diagnostic information and interfaces separately at the end of the section. We collect a list of potential questions the documentation should answer in Appendix A.

**change:** The running example builds around a black-box neural network trained on the three class version of the Abalone dataset (Nash et al., 1994). The network has a feed-forward architecture with four hidden layers, 32 neurons each, and ReLU activation functions. The final black-box validation accuracy is 0.64. The SToBB-artefact of our example comprises the information given after "**Example.**" in each subsection, together with supplementary material in Appendix B, the documented code base [2] and the therein contained answers to questions in Appendix A.

### 5.1   Observation Base

The first step is to set up the *observation base*, i.e., the structures that will store observations as they are collected over the black-box's life cycle. To implement this storage, one must specify the observation *space*, i.e., the relevant black-box input- and output variables and, where applicable, auxiliary measures derived from the black-box.

The documentation should explain why each variable or measure is included, how it is obtained, and what quality criteria apply. This enables third parties to assess whether the SToBB rests on a relevant and reliable empirical basis. As the black-box is used, new observations are recorded in the observation base and must be incorporated into the SToBB, expanding its empirical base.

**Example.**   The observation space consists of the input of the black-box (a 7 dimensional real-valued feature vector) and the corresponding class label predicted by the black-box. Further, each sample is associated with attribution scores computed with LIME (Ribeiro et al., 2016). The attribution scores act as auxiliary observables used to constrain the surrogate to subspaces marked as important to the black-box decision. The scores are obtained using the official LIME implementation[3], using default parameters of an exponential kernel with a bandwidth of 1.93 and sampling 5000 perturbations. An observation is rejected if all attribution scores are smaller than zero.

~~The observations are stored inside data fields of a "CoBoT class" object (see SECREFXX).~~

### 5.2   Hypothesis Class

The researcher needs to specify the hypothesis class of surrogate models, which determines the structures that operate on the observations. The documentation should describe how a surrogate processes an observation step by step, making the trace logic explicit, and specify conditions under which an observation is *not*

---

[2]Code available at anonymous.4open.science/r/cobot_tmlr-F583
[3]Version 0.2.0.1 from pypi.org/project/lime

covered. It should also record how user-centred criteria or other pragmatic virtues informed the design of the class and how these choices constrain its expressivity.

**Example.** As the black-box operates on tabular data, we choose a rule based hypothesis class, which is a popular format for explanation (Lakkaraju et al., 2016). Rules are represented as axis-aligned bounding boxes defined over subspaces of the input domain.

Formally, let $d \in \mathbb{N}$ be the input dimensionality and let $[1..d] := \{1, \ldots, d\}$. We define the hypothesis class as a *set of boxsystems*

$$\mathcal{B} \; = \; \big\{\, (I, B) \;\big|\; I \subseteq [1..d], \; B = \{b_1, \ldots, b_m\} \,\big\},$$

where each pair $(I, B)$ is called a *boxsystem*. Here, $I$ denotes a fixed subspace of input dimensions and $B$ is a finite set of *non-overlapping, axis-aligned, fully bounded boxes $b$* defined on that subspace. Each $b \in B$ is associated with a class label $c \in \mathrm{C}$.

The subspaces $I$ are derived from a local explainer $\Phi$ via an indicator function $\mathcal{I}$. Given an input sample $x$, $\Phi$ yields an attribution vector $a \in \mathbb{R}^d$, and

$$I \; = \; \mathcal{I}\big(a\big) \subseteq [1..d]$$

denotes the set of dimensions deemed relevant for the corresponding black-box prediction. For each distinct feature set $I$ encountered in the observation base, exactly one corresponding boxsystem $(I, B)$ is maintained.

All observations whose local explanations induce the same subspace $I$ are associated with the corresponding boxsystem and constrain its boxes. By construction, each observation is associated with exactly one subspace $I$ and, within the corresponding boxsystem, with at most one box $b$.

*Tracing logic of $\mathcal{B}$.* Given a sample $x$, let

$$I_x \; = \; \mathcal{I}\big(\Phi(x)\big) \quad \text{and} \quad c_x \; = \; f(x),$$

**change:** where $f : \mathbb{R}^d \to \mathrm{C}$ is the black-box function. Tracing succeeds if there exists a boxsystem $(I_x, B) \in \mathcal{B}$ and a box $b \in B$ such that                                                                                      GiiZ

$$x \in b \quad \text{and} \quad \text{LABEL}(b) = c_x.$$

If no such box exists, tracing fails and an update is triggered (see *update mechanism* Section 5.3).

*User-centric considerations.* The size of $I$ directly dictates the length or complexity of each rule, a factor that plays an important role in the user-friendliness of rules (Nauta et al., 2023; Ribeiro et al., 2018). To control this, we introduce a parameter $k \in \mathbb{N}$ that limits the number of selected dimensions.

For a given $k$, we define a constrained indicator function $\mathcal{I}^{k\uparrow}$ such that, for an attribution vector produced by $\Phi$,

$$\mathcal{I}^{k\uparrow}(a) \; \subseteq \; \{\, i \in [1..d] \mid a_i > 0 \,\}, \qquad |\mathcal{I}^{k\uparrow}(a)| \leq k,$$

where $\mathcal{I}^{k\uparrow}(a)$ contains the indices of the *highest-scoring positive* components of $a$.

~~While this does limit the expressive power of the hypothesis class, we describe in the following subsection a principled way by which the algorithm chooses the smallest value for $k$ that allows empirical adequacy, automating the trade-off between adequacy and user-friendliness.~~

## 5.3 Algorithmic Components

The algorithmic components define how a surrogate is *constructed* from a set of observations and how it is *updated* when adequacy fails. The description must make the procedure reproducible and should specify whether the algorithm can represent the full hypothesis class. The update procedure has to be described in equal detail, stating under what conditions updates are triggered, what operations may apply during updates, and how updates restore empirical adequacy.

**Example.** **change:** To obtain a model from the hypothesis class, we introduce the *Constructive Box Theoriser* (CoBoT) algorithm. For the sake of clarity and simplicity, we focus on describing **change:** relevant properties of the algorithm here, for a full technical description of CoBoT the reader is referred to Appendix B.1.

*Construction*: CoBoT incrementally builds the box system, refining it when new data points are added. The algorithm ensures that adequacy is always preserved while optimising for complexity. We initialise CoBoT with $k = 3$ and an empty set $\mathcal{B} = \emptyset$. Processing the observations one at a time, CoBoT incrementally adds boxsystems to $\mathcal{B}$. For each boxsystem the algorithm guarantees that 1) none of the boxes overlap, 2) each processed observation is unambiguously associated with one box, and 3) all samples associated with one box have the same label. The $k = 3$ dimensions used are the three dimensions with the highest attribution scores greater than zero.

*Update Mechanism*: Revision is conducted in case inconsistencies arise. These can occur in three ways:

1. When processing the local explanation: The local explanation indicates a subspace that has not been encountered before. For this subspace, a boxsystem is created, within which a single box, containing the new observation, is placed.

2. When processing the feature values: A boxsystem for the subspace exists, but within the subspace the sample lies outside of any box. The algorithm attempts to expand the boundaries of existing boxes (of the correct label) to include the point. If an expansion is found that leads to no inconsistencies with other observations, the algorithm stops. If no possible expansion is found, the sample is placed inside a new box containing only itself.

3. Misclassification: The observation is mapped to a box with a different class label than predicted by the black-box. The erroneous box is then dissolved, and its contained samples are merged into existing boxes whenever possible; otherwise, new boxes are created.

To illustrate the update mechanism, Figure 2 (Appendix B.2) shows several updates of the box-system for $I = \{2, 6\}$ until its current state.

**change:** [[Subspace size adaptation moved to appendix because not relevant to the example]]

### 5.4 Adequate Surrogate

Once the observation space, hypothesis class, and algorithmic components are defined, the researcher instantiates the first surrogate. The surrogate must be empirically adequate with respect to all observations available at this stage. If adequacy fails, adjustments to the algorithm or the hypothesis class may be required before continuing. After deployment, the surrogate must be maintained: when new observations arrive that break adequacy, an update must proceed via the documented update procedure.

**Example.** In our example, CoBoT instantiates the first surrogate by incrementally constructing $\mathcal{B}$ from the available observations, starting from $\mathcal{B} = \emptyset$. Adequacy is maintained incrementally by the revision operations described before.

### 5.5 Operation, distribution and maintenance

To be useful to parties other than its original creators, a SToBB must also be documented in terms of how it is operated and distributed. This includes specifying how the artefact is stored and versioned, and how external stakeholders can access it, e.g., via APIs or dedicated tooling. The documentation should outline the technical preconditions for using and updating the SToBB (such as required software, model access, or computational resources) and the processes that govern who may perform updates and under which conditions. Together, these provisions support structured use and maintenance of the SToBB by third parties over the life cycle of the black-box.

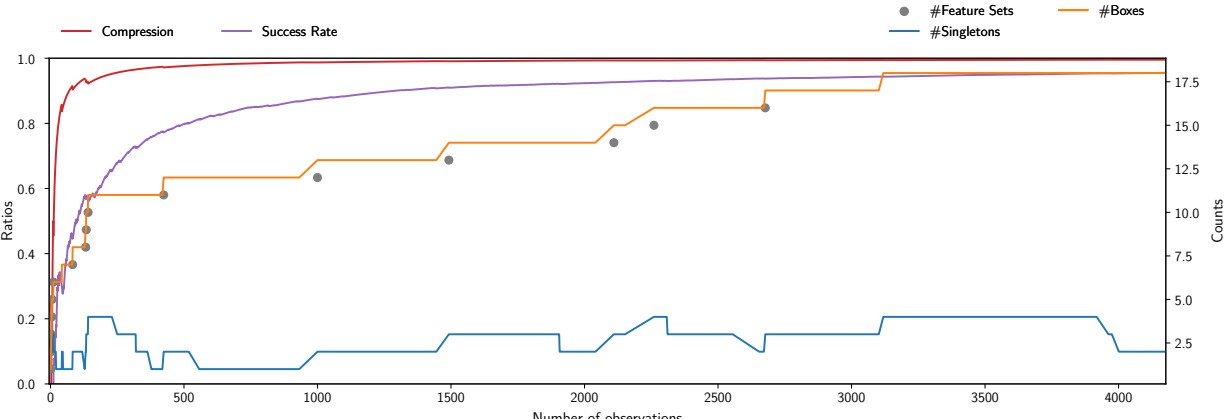

**Figure 1:** Diagnostic information on the current surrogate model. Left axis ("Ratios"): Compression is defined as the gain $1 - \frac{\#boxes}{\#samples}$; success rate is the fraction of samples that did *not* trigger an update. Right axis ("Counts"): #Feature Sets denotes the number of unique feature sets/distinct subspaces; #singletons and #boxes count the number of singleton and the total boxes, respectively. After processing all available 4 177 samples, the surrogate comprises less than 20 boxes. Across all observations, 187 updates were performed.

**Example.** As mentioned previously, interaction with the SToBB centres around a "CoBoT class" object. The object not only stores the observation base (Section 5.1), but also implements the algorithmic components (Section 5.3), contains the current surrogate (Section 5.4) and the record of its evolution as well as further diagnostic information (Section 5.6 below). The documentation of the code base contains the necessary information to load and interact with the provided object. The code base can be found at anonymous.4open.science/r/cobot_tmlr-F583 .

## 5.6 Optional: Diagnostic Information

After the adequate surrogate has been successfully instantiated, additional diagnostic information may be collected and documented, such as coverage statistics, structural complexity measures, or the number of updates applied. These measures do not test adequacy but complement the descriptive documentation with quantitative information.

**Example.** Each time CoBoT processes an update, we track internal statistics about outcome and state of the surrogate. We track the *total number of features sets*, the *total number of boxes* across all features sets, and the *total number of singletons* contained within those boxes. Figure 1 summarises the diagnostics over time for the current surrogate model. The current surrogate is based on 4 177 observations, of which 187 triggered updates (4.5%). Maximal subspace size remains as initialised, $k = 3$. The boxsystems of the surrogate occupy 16 distinct subspaces and contain a cumulative sum of 20 individual boxes. Two of those boxes are singletons, containing only a single sample. This yields a compression gain of $1-(20/4177) \approx 0.995$.

## 5.7 Optional: Interfaces

As discussed in Section 4.1, interfaces are not part of the SToBB itself but may be provided and documented to better support downstream use.

**Example.** We include example interfaces answering three common questions in XAI that can be attributed to the leading questions *Why*, *How to be that*, and *How*, respectively, as presented in Liao et al. (2020):

- **Local interface:** *Q: Why is this input given that prediction?* What features were important? Given an input, the interface returns a rule by extracting the box constraints relevant for that

instance. This query can trigger an update on the surrogate. A formal listing is given in Algorithm 2 (Appendix B.3).

- **Contrastive interface:** *Q: Why is this input given this prediction and to what values would the important features need to change to lead to a different outcome?* For a given input, the interface extracts the applicable box and the closest box with a different label from the same boxsystem. This query can trigger an update on the surrogate. A formal listing can be found in Algorithm 3 (Appendix B.3).

- **Global interface:** *Q: What subspaces are considered important and how do they vary with class label and regions of the input space?* To answer this question, the interface first computes a 2D-UMAP projection of all input-coordinates in the observation basis. Each sample is then assigned to its corresponding box and coloured according to the respective subspace (important dimensions), while class labels are encoded by marker shape. This provides a visual account of how the importance of different dimensions changes over the data manifold. This interface cannot trigger updates. A listing of the procedure and its output are provided in Algorithm 4 and Figure 4, respectively (Appendix B.3).

## 6 Discussion and Conclusion

We introduced the notion of a *scientific theory of a black box* (SToBB), a structured and auditable artefact that consolidates explanatory information about a fixed machine learning model across its life cycle. Building on Constructive Empiricism, we identified three obligations that guide its design: empirical adequacy with respect to observed behaviour, adaptability to new observations, and auditability through documentation. By separating the persistent representational foundation of explanation from the context-dependent methods that present answers to stakeholders, the SToBB framework addresses a specific and so far unoccupied role in XAI: a life cycle-scale, inspectable point of reference that supports consistent, reusable analyses and systematic external scrutiny. The following subsections discuss the conceptual positioning of this contribution, its current limitations, and directions for future work.

### 6.1 Discussion

Following existing XAI taxonomies (Speith, 2022), a SToBB can be described as a global, post-hoc artefact that centres a surrogate model. This characterisation, however, neither captures the temporal scope of the SToBB perspective nor the particular CE-imposed obligations the surrogate has to fulfil. More importantly, it misses what the SToBB perspective adds beyond a surrogate description: it makes explicit a maintained representational basis that supports a clear distinction between *explanatory information* and *explanation*. The explanatory information, represented by the observation base and the surrogate, can be designed and evaluated primarily with respect to empirical adequacy and traceability, independently of the interfaces that must take user-centred qualities into account. The resulting baseline guarantees give interface designers an explicit point of reference for making principled trade-offs between information richness and the users' need for comprehensible answers, a well-known pain point in XAI research (Jacovi & Goldberg, 2020; Doshi-Velez & Kim, 2017). As illustrated by the CoBoT example, a SToBB may still be constructed with user-centred criteria in mind, but whenever there is tension, empirical adequacy takes precedence.

The hypothesis class of a SToBB naturally induces a language bias that favours some types of explanations over others. In our example, we used the black-box's application domain to deliberately target rule-based explanations, which motivated choosing axis-aligned boxes as the surrogate structure. As we also saw, this choice doesn't strictly limit interfaces to only produce this one type of explanation. Interfaces are free to transform the information contained in the SToBB as the context requires. Two directions for further exploration seem particularly natural: (i) developing other hypothesis classes and algorithmic components directly derived from established interpretable model families, such as linear or logistic regression and prototype-based models (Molnar, 2020); and (ii) exploring existing explanation methods to assess whether and how they can be adapted into SToBB components. For methods such as Anchors (Ribeiro et al., 2018), one

could imagine SToBB variants whose hypothesis class consists of sets of local rules, with algorithmic components that generalise from individual anchors and update them as new observations arrive. Working out such adaptations requires rethinking these methods as parts of a cumulative and updateable process rather than as single-shot explainers. The work of Naik & Turán may provide a useful starting point for interface design that complements the SToBB framework particularly well. Where our paper transfers van Fraassen's Constructive Empiricism to XAI, Naik & Turán (2020) transfer van Fraassen's *theory of explanation* to XAI that guides the formalisation of questions and answers.

In this work we treated the black-box as fixed: once trained, its input-output behaviour does not change, i.e., all collected observations remain valid for the lifetime of the SToBB. In practice, an AI system may make use of different black-box versions over its life cycle, e.g., through fine-tuning or re-training. **change:** This is a consequence of model change rather than a limitation of the SToBB framework itself: any explanatory artefact tied to a specific model instance faces the same constraint. When the model is replaced, a new SToBB needs to be instantiated for the updated instance. Crucially, the accumulated observation base is thereby not useless: while predictions and auxiliary measures may become stale, the input samples remain useful as probes of regions of the input space that were relevant during the previous model's lifetime and can therefore serve as a principled starting point for rebuilding the observation base of the new SToBB. Developing efficient reconstruction procedures and principled criteria for when inherited observations must be revalidated is left for future work. GiiZ

Specifically designed to be empirically adequate and extensible, SToBBs address use-cases that differ from those of many existing XAI methods. Their focus on accumulating information during the life cycle of a black-box positions them primarily as tools for AI governance. At a minimum, a SToBB presupposes stable access to a fixed model instance whose input-output behaviour can be logged, the ability to store and process observations in compliance with applicable legal and organisational rules, and an interpretable hypothesis class and update procedure that relevant stakeholders can inspect. Using SToBBs in practice will require decisions about how to regulate access, updates, and versioning, e.g., who may read or modify a SToBB, how documentation integrity is ensured over time, and how privacy and regulatory constraints affect what observations can be stored and which parties may query which parts of the artefact. A SToBB provides a structured target for such access within the bounds set by the surrounding governance regime, but it cannot compensate for missing access to model behaviour or override constraints on data retention and sharing. **added:** The latter point reflects a broader tension in trustworthy ML between transparency and data minimisation: accumulating observations for auditability and limiting the accumulation of personal data beyond necessity — as required, for example, by data protection regulations such as the GDPR — pull in opposite directions, and the appropriate balance will depend on the deployment context. The concrete answers to these governance questions lie beyond our discussion here but are of practical relevance to the adoption of SToBBs. CQYP

## 6.2 **added: Limitations**

The following points identify specific constraints on the current framework and its illustration that future work will need to address.

This paper focuses on the conceptual development of a SToBB; the accompanying illustration is necessarily limited in its scope. The CoBoT example targets one data-domain, incorporates a single auxiliary measure, and provides an online learner that covers construction and update in one procedure. Because the SToBB framework formulates only abstract, qualitative constraints, it leaves considerable freedom in how auxiliary measures, hypothesis classes, and algorithmic procedures are designed. **added:** Claims about lifecycle-scale utility, stakeholder reuse, and extensibility follow from the framework's design obligations rather than from empirical demonstration, and should be understood as motivated conjectures that future instantiations will need to examine. CQYP

**added:** The observation space may include auxiliary measures derived from the black box, such as the feature attributions used in our illustration, that are subject to instability. CE treats observations as potentially erroneous and permits their rejection, which provides a principled starting point: an observation whose auxiliary measure is deemed unreliable can be excluded from the observation base. However, when instability CQYP, aMTd

is systematic rather than occasional, the surrogate structure derived from the auxiliary measure may be compromised in ways that individual rejection criteria cannot fully address. The documentation obligation of a SToBB requires the choice of auxiliary measures to be made explicit and its quality criteria to be stated, which at minimum makes this source of fragility inspectable by third parties. Best practices for auxiliary measure selection and quality assessment will develop as SToBBs are instantiated across different settings and hypothesis classes.

**added:** The observation base grows as the black-box is used, and empirical adequacy as defined requires the surrogate to remain consistent with all recorded observations. CE does not address the deletion of observations, creating tension with data retention constraints that may apply in some deployment contexts, a concern shared by any approach that accumulates input-output behaviour for auditing purposes. One direction could be to identify regions of the input space where the surrogate structure has been stable over time and where observations could potentially be summarised or pruned without violating adequacy on active regions. Strategies for observation base management and their implications for maintained adequacy are left for future work.

GiiZ

### 6.3 Conclusion

**change:** Future work will need to explore the framework across a broader range of domains, hypothesis classes, and auxiliary measures, develop principled interfaces for diverse stakeholder needs, and address the practical aspects of deployment, maintenance, and governance that concrete adoption will raise. Over time, instantiations of different shapes can be explored and best practices established. The SToBB framework opens a direction for XAI research that goes beyond the aim of producing better individual explanations and takes on the challenge of building a trustworthy, auditable foundation from which good explanations can be consistently derived.

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

# A  Example Documentation Questions

The goal of the documentation is to maximize transparency and informed use of third parties. The questions listed below may serve as a starting point:

---

**Scientific Theory of a Black Box (SToBB) – Documentation Question Sheet**

**Observations**
- What variables, auxiliary measures and target outputs constitute the observation space?
- How are auxiliary measures of the black-box behaviour obtained?
- What quality criteria apply to these measurements?
- How is the observation base stored?

**Hypothesis Class**
- What interpretable model family defines the surrogate's relational structure?
- How does the surrogate process an observation step by step?
- Under which conditions is an observation not covered by the surrogate?
- Which user-centred criteria or pragmatic virtues shaped the design?

**Algorithmic Components**
- How is the surrogate initially constructed from the observation base?
- How does the surrogate demonstrate empirical adequacy?
- How is the surrogate stored?
- What is the update policy when new observations appear?

- Does the algorithm represent the full hypothesis class?
- How are trade-offs or approximations documented?

**Operational Requirements**
- What resources or runtime requirements must be met to deploy and maintain the SToBB?
- Does access to parts of the SToBB have to be restricted for specific users or interfaces?
- How are updates to the SToBB performed?
- How are updates to the SToBB recorded?

**Diagnostics (if any)**
- Which diagnostic metrics are monitored during construction and updates?
- How frequently are diagnostics evaluated and recorded?
- How are diagnostic records stored and made accessible?

**Interfaces (if available)**
- What interfaces are implemented and what questions do they answer?
- Who is the target audience of the interface?
- What is the purpose, method, and assumption of each interface?
- How can external parties audit or replicate the analyses?

---

The CoBoT repository anonymous.4open.science/r/cobot_tmlr-F583 contains answers to these questions for our proof-of-concept.

# B Extended documentation for the example SToBB

This section of the appendix contains a detailed description of the CoBoT algorithm, more detailed diagnostic information from the concrete example and a formal description of the interface functions.

## B.1 Algorithmic Component CoBoT — Step-by-Step Description

Following the brief description of CoBoT in Section 5.3, we now give a detailed description of the algorithm, going through Algorithm 1 step by step.

---

**Algorithm 1** Construction and Update procedure of Constructive Box Theoriser (CoBoT)

---

**Input:** Black-box model $f : \mathbb{R}^d \to$ C for some positive integer $d$ and finite set C, local explainer $\Phi$ for $f$, a binarization function $\mathcal{I}^{k\uparrow}$ for local explanation and $k$, input sample $x$, set of boxsystems $\mathcal{B}$, set of encountered observations $\mathcal{O}$

**Output**: Set of boxsystems $\mathcal{B}$

1: $c \leftarrow f(x)$
2: $a \leftarrow \Phi(f, x, c)$
3: $I_x \leftarrow \mathcal{I}(a)$         ▷ obtain set of important dimensions
4: $\mathcal{O} \leftarrow \mathcal{O} \cup \{(x, c, I_x)\}$
5: $B_{I_x} \leftarrow B$ **if** $\exists (I, B) \in \mathcal{B}$ with $I = I_x$, **else** None
6: **if** $B_{I_x} =$ None **then**         ▷ None on first encounter of itemset
7:     $b_x \leftarrow \text{CREATESINGLETON}(x, c, I_x)$
8:     $\mathcal{B} \leftarrow \mathcal{B} \cup \{(I_x, \{b_x\})\}$
9: **else**
10:     **if** $\exists b \in B_{I_x}$ with $x \in b$ **then**
11:         **if** $\text{LABEL}(b) \neq c$ **then**
12:             $B_{\mathcal{O}_b} \leftarrow \{\text{CREATESINGLETON}(o, c_o, I_o) \mid (o, c_o, I_o) \in \mathcal{O} \wedge I_o = I_x \wedge o \in b\}$
13:             $B'_{I_x} \leftarrow \text{MERGE}((B_{I_x} \setminus \{b\}) \cup B_{\mathcal{O}_b})$
14:             $\mathcal{B} \leftarrow (\mathcal{B} \setminus \{(I_x, B_{I_x})\}) \cup \{(I_x, B'_{I_x})\}$
15:     **else**         ▷ no existing box contains sample
16:         $b_x \leftarrow \text{CREATESINGLETON}(x, c, I_x)$
17:         $B'_{I_x} \leftarrow \text{MERGE}(B_{I_x} \cup \{b_x\})$
18:         $\mathcal{B} \leftarrow (\mathcal{B} \setminus \{(I_x, B_{I_x})\}) \cup \{(I_x, B'_{I_x})\}$
19: **return** $\mathcal{B}, \mathcal{O}$

---

*1) Initialization* As input, CoBoT requires a black-box model $f$, an input sample $x$, a local explainer function $\Phi$, an indicator function $\mathcal{I}^{k\uparrow}$, a list of observations $\mathcal{O}$, and a (possibly empty) set of boxsystems $\mathcal{B}$. The indicator function $\mathcal{I}^{k\uparrow}$ transforms a local explanation into a set of indices $I$ by extracting the top $k$ most important feature dimensions. In lines 1 to 3, CoBoT computes the class label $c$ for the input sample $x$ and the local explanation $a$, from which it derives the feature set $I_x$ of most important dimensions. The observation triple $(x, c, I_x)$ is then added to the set of observations $\mathcal{O}$ in line 4.

*2) Creating a new boxsystem* Line 6 begins with the consistency check and the updating logic. It checks whether the feature set $I_x$ is present in the set of boxsystems. If not, line 7 creates a singleton-box $b_x$ around $x$ in the subspace determined by $I_x$, associates it with the corresponding class label, and adds the tuple $(I_x, \{b_x\})$ to $\mathcal{B}$. CoBoT then returns the updated set of boxsystems $\mathcal{B}$ and observations $\mathcal{O}$.

*3) Resolving inconsistency* In case $I_x$ had been encountered before, CoBoT proceeds in line 10 and checks if there exists a box $b$ in the corresponding boxsystem $B_{I_x}$ that covers $x$. If so, line 11 checks if the box's label matches the black-box output $c$. If the labels match, the algorithm returns. Otherwise, the boxsystem is updated: Line 12 selects all samples from the current list of observations that lie within the affected box and places each into its own singleton box. In the next step (line 13), the inconsistent box $b$ is removed from $B_{I_x}$ and all singleton-boxes in $B_{\mathcal{O}_b}$ are attempted to be *merged with* the remaining boxes in $B_{I_x}$. To this end, we use an adaptation of the algorithm presented in Stadtländer et al. (2024). Given a set of boxes,

the algorithm greedily selects the closest two boxes having the same class, and attempts to join them. If a join leads to an overlap with any other box, the join operation is reverted. This process continues until no further consistent mergers are possible. While the original algorithm by Stadtländer et al. was developed for binary classifications, we adapt it mutatis mutandis to multi-class problems. The algorithm has several desirable properties for our scenario:

i It naturally guarantees that *all and only indicated* dimensions in $I_x$ are used. This guarantees full control over rule complexity.

ii The algorithm guarantees to return a boxsystem that is *fully* consistent with the training data, thereby fulfilling empirical adequacy.

iii Boxes are unambiguous: None of the computed boxes overlap, meaning that 1) each sample induces the construction of at most one box and 2) no sample can fall into two boxes at the same time.

iv All boxes are bounded on all sides and do not extend beyond values observed in the input data. Thus, each box is strictly tied to observations.

If boxes from two different boxsystems, say $B_{I_1}$ and $B_{I_2}$, overlap, this is of no concern, because $\Phi$ indicated that $f$ used different sets of features for its decision. Continuing on line 15, the set of boxsystems is updated with the adapted boxsystem and CoBoT returns.

*4) Expanding coverage*  The final *else* case is entered if a boxsystem does exist for $I_x$, but none of the boxes in $B_{I_x}$ cover the sample. In that case line 16 creates a new singleton box around $x$ and line 17 performs the merge operation described above in an attempt to extend any existing consistent box by the singleton-box. The set of boxsystems is updated and CoBoT returns.

moved from main part:
*Adapting maximal rule-complexity parameter $k$*: If an incoming observation shares the same important subspace and identical features values with a previous observations, yet is associated with a different label, CoBoT returns an error. In this case, the two observations are indistinguishable in the selected subspace. However, since the black-box assigns them different values, this indicates that either (i) the dimensionality of the subspace is too small (i.e., the pragmatic requirement was too strict), or (ii) the feature attribution method is not working as intended. To address case (i), CoBoT increases $k$ by one and rebuilds the surrogate from scratch using all stored observations, while recording its previous state and the conflicting sample for later inspection. Case (ii) occurs if CoBoT fails to build an empirically adequate $\mathcal{B}$ despite using all important dimensions in an observation. ~~In this case, the observation space is inherently ambiguous and has to be modified externally, e.g., by the maintainers of the SToBB that then need to rebuild the observation base.~~

## B.2  Further Diagnostic Information

This further complements the diagnostic information shown in the curves in Figure 1, Section 5.6.

**Increments of maximal subspace size**  CoBoT was initialized with $k = 3$, this has not been increased.

**Evolution of a box-system**  Despite the current subspace size being $k = 3$, the surrogate uses subspaces with less than 3 dimensions. Figure 2 shows the subspace of the boxsystem for feature set $I = \{2, 6\}$ at different points in time. Two classes are present, indicated by the two different box colours. Samples of individual observations are visualized as dots. The scatterplot indicates that the features are linearly correlated within the subspace. Of the 187 total updates to the surrogate CoBoT performed, 27 were performed on this boxsystem.

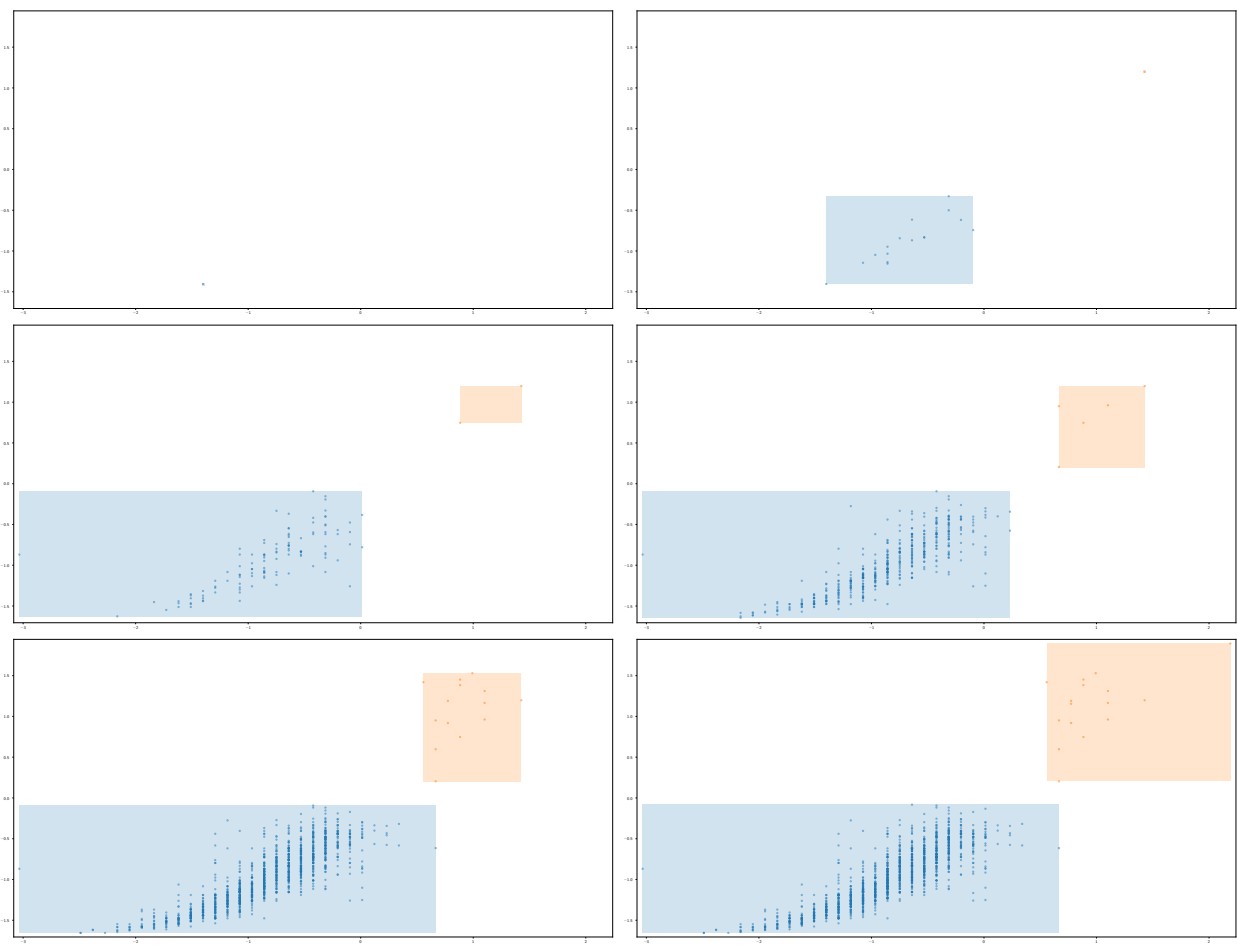

**Figure 2:** Evolution of the axis-aligned bounding boxes contained in the boxsystem for subapace $I = \{2, 6\}$. Each individual image shows the same boxystem after different updates. Updates number [1, 7, 16, 19, 25, 27 (final)]. Colors encode classes.

## B.3  Interfaces

We provide interfaces addressing categories of questions listed in the XAI question bank (Liao et al., 2020). By design, the surrogate's hypothesis class naturally supports extracting rule based explanations. Algorithm 2 and Algorithm 3 extract a local and a contrastive explanation, respectively. Algorithm 4 gives a global view.

Algorithm 2 answers the question "Given $x$, why did $f$ predict $c$?" by returning the applicable box. By instead returning the supporting samples of the relevant box, a different interface could give an "explanation by example", answering the question "What kind of instance gets the same prediction" . Algorithm 3 answers the question "Given $x$, why did $f$ predict $c$ and what other class is the sample most similar to?", supplying in its output also a neighboring box of a different class. It could be adapted to give a targeted answer "Why $c$ and not $c'$ ?" by restricting the search in line 9 to $b$ associated with $c'$. In case the request from any of the interfaces leads to a failure condition (*itemset unknown, not covered, wrong label*), CoBoT automatically updates the surrogate. **added:** Example outputs are listed in Figure 3.

---

**Algorithm 2** LOCAL Explanation

**Input:** $\mathcal{B}$, $I_x$, $c$, $x$

1: $B_{I_x} \leftarrow B : ((I, B) \in \mathcal{B} \wedge I = I_x)$ **else** None
2: **if** $B_{I_x} \neq$ None **then**
3:   $b_x \leftarrow b : b \in B_{I_x} : x \in b$ **else** None
4:   **if** $b_x \neq$ None $\wedge$ LABEL$(b_x) = c$ **then**:
5:     **return** $b_x$
6: UPDATESURROGATE$(x)$    ▷ Call CoBoT's update mechanism to integrate $x$ in $\mathcal{B}$
7: LOCAL$(I_x, c, x, \mathcal{B})$

---

**Algorithm 3** CONTRASTIVE Explanation

**Input:** $\mathcal{B}$, $I_x$, $c$, $x$, a distance function $d(.,.)$ between two boxes

1: $B_{I_x} \leftarrow B : ((I, B) \in \mathcal{B} \wedge I = I_x)$ **else** None
2: **if** $B_{I_x} \neq$ None **then**
3:   $b_x \leftarrow b : b \in B_{I_x} : x \in b$ **else** None
4:   **if** $b_x \neq$ None $\wedge$ LABEL$(b_x) = c$ **then**:
5:     $b_{\neg c} \leftarrow \underset{b' \in B_{I_x} \wedge \text{LABEL}(b) \neq c}{\text{argmin}} \; d(b_x, b')$
6:     **return** $(b_x, b_{\neg c})$
7: UPDATESURROGATE$(x)$
8: CONTRASTIVE$(\mathcal{B}, I_x, c, x, d)$

---

To obtain a global view on $\mathcal{B}$, we define Algorithm 4. It computes a 2d UMAP embedding of all observations, color-coded by the feature set they were mapped to and the marker shape indicating the class label predicted by $f$. Figure 4 shows the interface outputs for the current set of the SToBB. The color imbalance highlights that the feature sets have vastly different support. Feature set $\{2, 6\}$ (light pink) occupies most of the right. Despite the comparatively large support of the feature set, the number of updates it triggered was below 15% of all updates (27/187). The left side of the plot is occupied by more feature sets, brown, blue and green. Purple, lavender, cyan and orange have very little support and are confined to very small areas. Colours form perceptibly connected regions. Conspicuously, many feature sets seem to use only a single type of marker, indicating that classes are separable not only by value but by feature alone. More detailed statistics could be delivered by another interface.

---

**Algorithm 4** Global Visualization

**Input:** Set of observations $\mathcal{O} = \{(x, c_x, I_x)\}$, a 2D projection operator UMAP$(\cdot)$
**Output**: 2D visualization of projected samples colored by $I_x$ and shaped by $c_x$

1: $X \leftarrow \{x \mid (x, c_x, I_x) \in \mathcal{O}\}$
2: $(u_x, v_x)_{x \in X} \leftarrow$ UMAP$(X)$
3: Visualization $\leftarrow$ EMPTYPLOT
4: **for** each $(x, c_x, I_x) \in \mathcal{O}$ **do**
5:   color$_x \leftarrow$SUBSPACETOCOLOR$(I_x)$     ▷ color encodes important dimensions
6:   marker$_x \leftarrow$CLASSTOMARKER$(c_x)$     ▷ marker encodes class label
7:   Visualization.ADDSCATTER$((u_x, v_x), \text{color}_x, \text{marker}_x)$
8: **return** Visualization

---

**Local interface output**

```
feature names [Length, Diameter, Height, Whole_weight, Shuckes_weight, Viscera_weight, Rings]

sample [95.16, 81.09, 26.88, 133.77, 66.20, 30.79, 32.08 ]

predicted class 0 (Young) in box
   [Height in [0.00, 34.00] and Rings in [1.00, 45.00]]
```

aMTd

**Contrastive interface output**

```
feature names [Length, Diameter, Height, Whole_weight, Shuckes_weight, Viscera_weight, Rings]

sample [90.79, 59.80, 23.00, 77.31, 42.57, 23.89, 24.48]

predicted class 0 (Young) contained in box
   [Height in [0.00, 34.00] and Rings in [1.00, 48.10]]
closest other class is class 1 (Intermediate) with box
   [Height in [33.00, 48.00] and Rings in [53.00, 100.30]]
```

**Figure 3:** **change:** Example outputs of the local (top) and contrastive (bottom) interfaces querying the maintained SToBB.

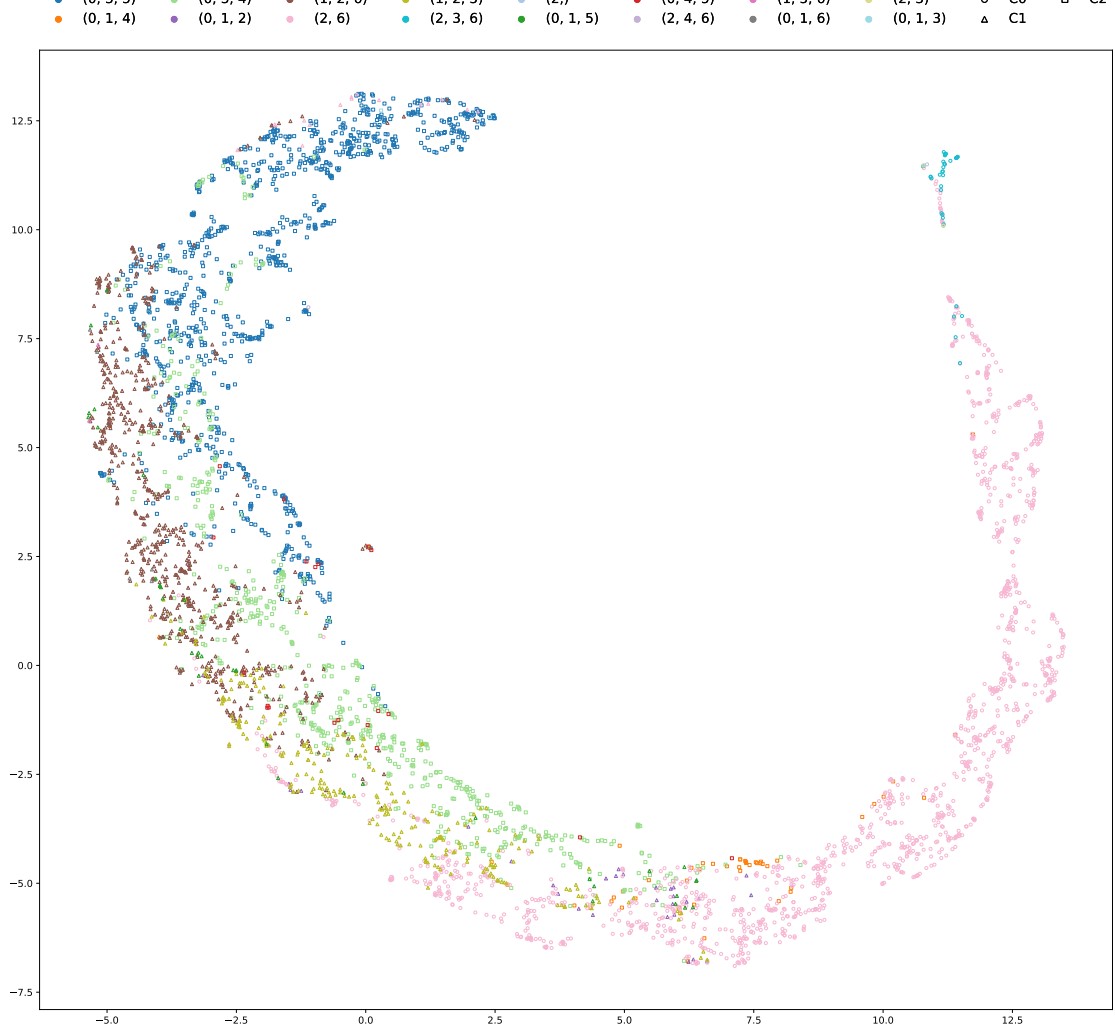

**Figure 4:** 2d UMAP embedding of the 4 177 observed samples in the CoBoT-SToBB. Colours indicate feature sets from $\mathcal{I}^{3\uparrow}$, marker style indicates class label.

# Comparison Decision Tree and VFDR baselines – NOT PART OF THE PAPER (?)

This section is not part of the paper and is provided for the reviewers' reference only.

To address concerns regarding CoBoT's structural growth behaviour and to provide a point of reference against existing rule-based approaches, we compare CoBoT against a decision tree baseline and VFDR [vf] across four datasets of varying dimensionality and size (see Table 2). This is not intended as an exhaustive benchmark but as a contextualisation of CoBoT's behaviour under the adequacy constraint.

CQYP,
GiiZ,
aMTd

| Dataset | Classes | Features | Samples | Black-box accuracy |
|---|---|---|---|---|
| Abalone | 3 | 7 | 4177 | 0.64 |
| Beans [be] | 7 | 16 | 13611 | 0.92 |
| Breastcancer [bc] | 2 | 30 | 569 | 0.96 |
| Spambase [sp] | 2 | 53 | 4601 (4548) | 0.94 |

**Table 2:** Relative to Abalone, Beans is higher dimensional and provides significantly more data. Breastcancer provides very little data while still increasing dimensionality. Spambase has comparable amounts of data, but increases the dimensionality most drastically. In case of Spambase 53 observations were excluded where feature attributions contained negative values only.

**Decision tree baseline** refits a decision tree classifier on all observations seen so far each time a new observation is not covered by any existing leaf, mirroring CoBoT's update trigger. The tree is unconstrained, i.e., it is allowed to grow without regularisation, so that empirical adequacy is achievable by construction on the same terms as CoBoT. Since the tree is refit from scratch on all accumulated observations at each update, its structure is not forced to monotonically grow; a new fit may produce a more compact tree. Leafs that contain a single observation are counted as singletons. We also track the maximum tree depth. This has no direct equivalent in CoBoT, because CoBoT's maximum rule length is fixed by the subspace size. For reference we compare the tree depths against the number of different subspaces CoBoT covers.

**VFDR** is a natural point of reference as an incremental rule learner for streaming data, but its intended behaviour differs from CoBoT in two important respects. First, VFDR updates its rule statistics on every observation regardless of whether the current model covers it correctly; CoBoT only revises when an observation is not correctly covered. Second, VFDR does not enforce geometric non-overlap between rules and has no empirical adequacy guarantee: observations not covered by any explicit rule are absorbed by an undifferentiated *default-rule* with no revision trigger or documentation. We track the default-rule hit rate as a diagnostic of this behaviour. We run two configurations: VFDR [4] with default parameters except majority class prediction in rule leafs rather than Naive Bayes ("VFDR default"), and a permissive variant with Hoeffding bound conditions removed (expansion confidence and tie threshold set to 1.0), providing an upper bound on rule generation ("VFDR permissive").

**Setup and general figure layout** In each scenario we run 30[5] independent random permutations of all observations. The general figure layout is as follows:

- *Colouring:* CoBoT (blue), Decision Tree (orange), VFDR default (green), VFDR permissive (purple). All curves show the mean (line) and standard deviation (shaded) .
- *Top left:* Success rate (fraction of observations not triggering an update) and, for VFDR, post-update accuracy (dotted).
- *Top right:* Cumulative updates.
- *Bottom left:* Total boxes, leaves, and rules (solid); singletons for CoBoT and the decision tree (dashed); VFDR fraction of samples mapped to default-rule at time step (dotted, right axis, sliding window mean over 50 samples).
- *Bottom right:* Number of distinct CoBoT feature sets (left axis), decision tree maximum depth, and VFDR maximum rule length (right axis). Note that the max-length of a CoBoT "decision path" is 3 in all cases.

---

[4]Implementation used: SCIKIT-MULTIFLOW.VERYFASTDECISIONRULESCLASSIFIER

[5]Running VFDR is runtime intensive (particularly Beans); we can increase the number of random permutations upon request.

**Summary and discussion** Figures 5 to 8 visualise results for Abalone, Beans, Breastcancer and Spambase, respectively; result descriptions are provided in the captions. Across the four datasets, the results show that CoBoT's structural behaviour is task-dependent and sensitive to the ratio of observations to input dimensionality. Overall, CoBoT's structural growth, singleton count and compression behavior remains well behaved on all datasets, although convergence is not visible in the most extreme setting (Spambase, 53 features).

On Abalone and Beans, CoBoT achieves compact, stable surrogates with near-zero singletons, demonstrating that empirical adequacy is not achieved through fragmentation, indicating genuine generalisation. The comparatively larger complexity of the unconstrained decision tree shows that this compactness is not a trivial consequence of the task. On Breastcancer, CoBoT's box count stabilises despite the small sample size, whereas the decision tree's leaf count does not indicate to slow down. On Spambase, no algorithm converges, though CoBoT's complexity grows notably faster. This suggests that the difficulty is a property of the task rather than a specific limitation of CoBoT's design. VFDR's post-update accuracy never fulfills the empricial adequacy requirement, even in the permissive setting. The fluctuations of default-rule hit indicate that previously covered observations are reassigned to the default-rule.

Together, these results demonstrate that the adequacy requirement produces meaningfully different surrogate behaviour: across the datasets where the observation base is sufficiently dense, CoBoT achieves more compact surrogates than the decision tree with fewer singletons, while VFDR silently absorbs uncovered observations into a default rule without a revision trigger.

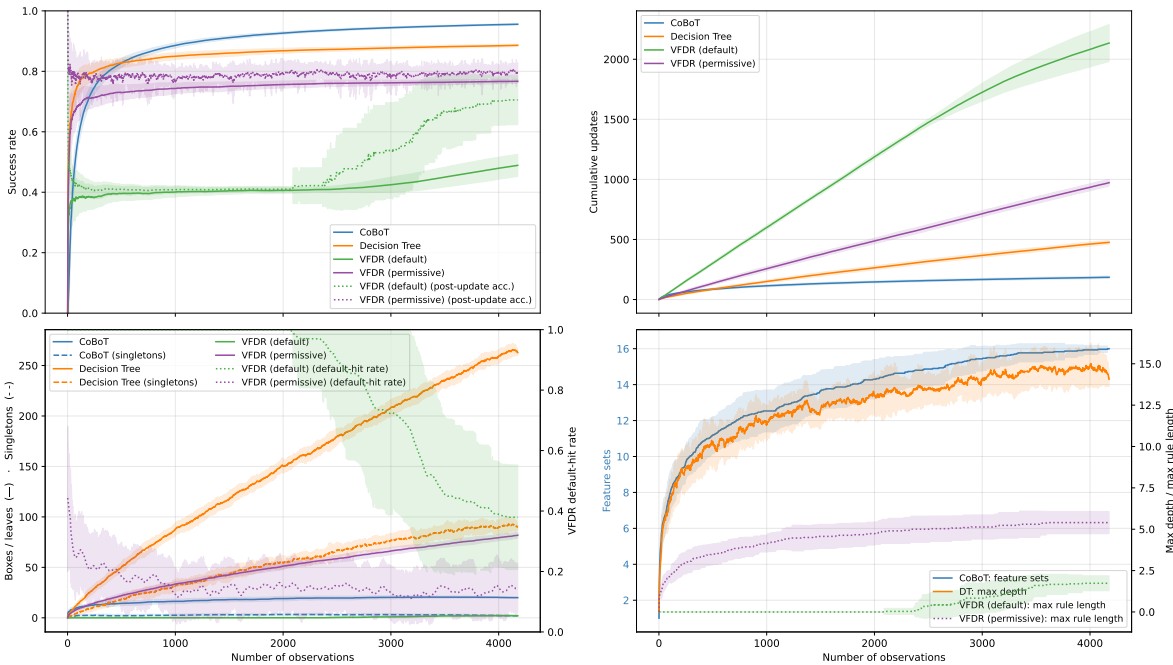

**Figure 5: Abalone** (3 classes, 7 features, 4177 samples, black-box accuracy 0.64). CoBoT reaches a success rate of approximately 0.95 with around 20 boxes and near-zero singletons. The decision tree plateaus at a success rate of approximately 0.88 with around 260 leaves and 90 singletons, triggering more than twice as many updates. VFDR default reaches a success rate of approximately 0.40 with almost no explicit rules and a default-rule hit rate near 1.0 for half of the sequence; VFDR permissive reaches a success rate of approximately 0.78 with a default-rule hit rate around 0.15.

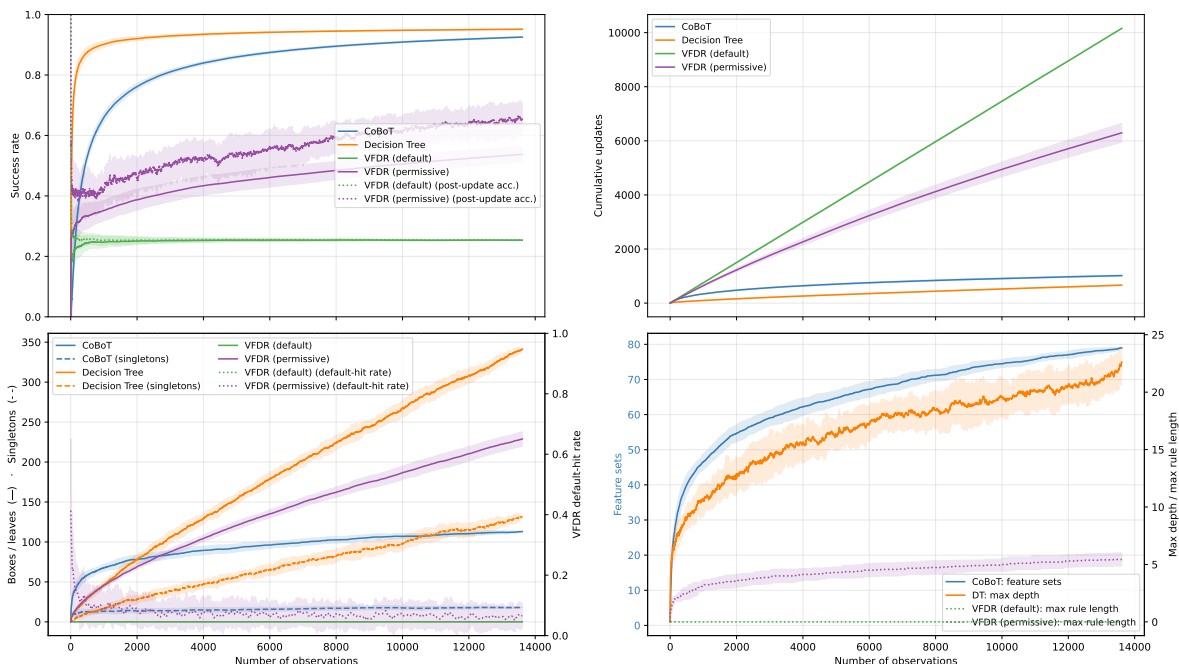

**Figure 6: Beans** (7 classes, 16 features, 13611 samples, black-box accuracy 0.92). CoBoT reaches a success rate of approximately 0.92 with around 110 boxes and around 30 singletons. The decision tree reaches a success rate of approximately 0.95 with around 340 leaves and 130 singletons. VFDR default reaches a success rate of approximately 0.25 with almost no explicit rules and a default-rule hit rate of 1.0 throughout; VFDR permissive reaches a success rate of above 0.60 with a default-rule hit rate around 0.05.

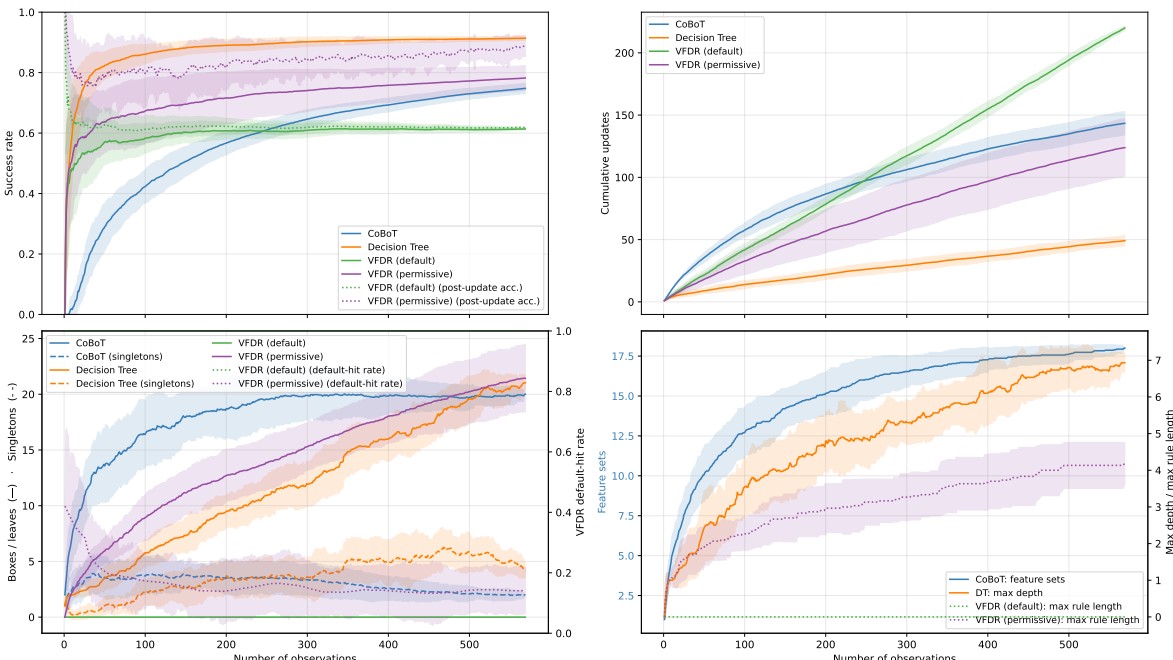

**Figure 7: Breastcancer** (2 classes, 30 features, 569 samples, black-box accuracy 0.96). CoBoT reaches a success rate of approximately 0.75 with around 20 boxes and approximately 2 singletons at the end of the sequence. The decision tree reaches a success rate of approximately 0.90 with around 20 leaves and 5 singletons. VFDR default remains at a success rate of approximately 0.60 with near-zero explicit rules and a default-rule hit rate near 1.0 throughout; VFDR permissive reaches a success rate of approximately 0.78 with a default-rule hit rate around 0.10.

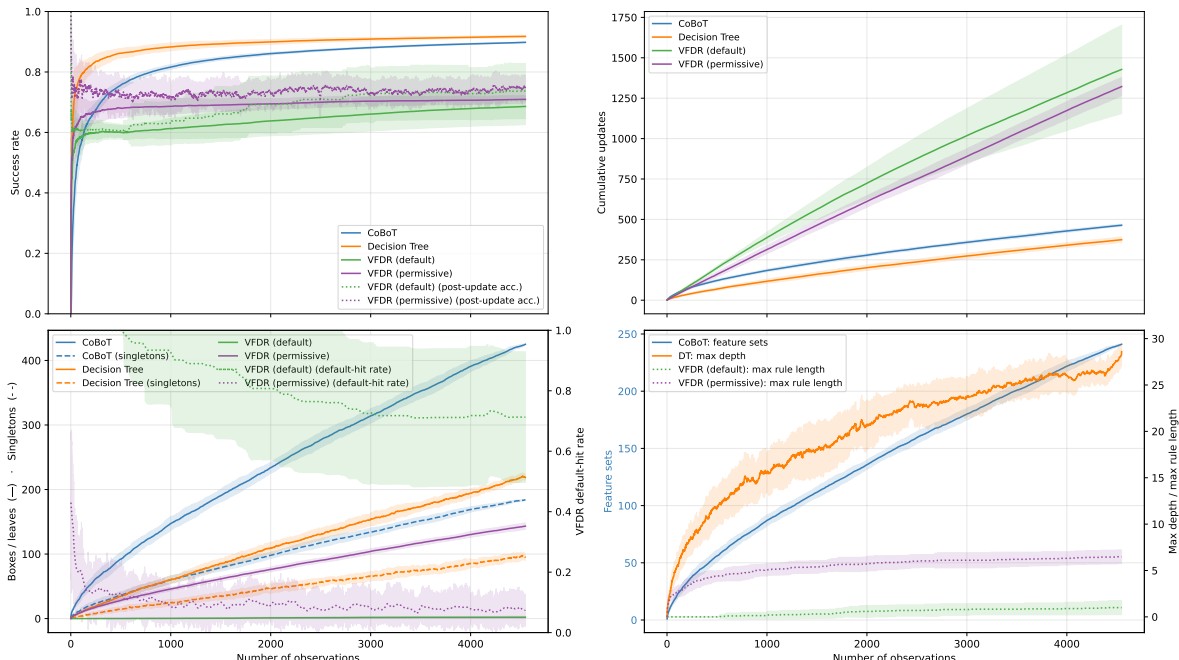

**Figure 8: Spambase** (2 classes, 53 features, 4548 samples, black-box accuracy 0.94). CoBoT reaches a success rate of approximately 0.85 with around 430 boxes and approximately 190 singletons, with both counts still growing at the end of the sequence. The decision tree reaches a success rate of approximately 0.90 with around 210 leaves and 100 singletons, also still growing. VFDR default reaches a success rate of approximately 0.65 with near-zero explicit rules and a default-rule hit rate between 1.0-0.75 throughout; VFDR permissive reaches a success rate of approximately 0.70 with default-rule hit rate below 0.1.

## Additional references

[vf] Petr Kosina and João Gama. Very fast decision rules for classification in data streams. *Data Min. Knowl. Discov. 29*, 1 (January 2015), 168-202, (2015) 10.1007/s10618-013-0340-z

[be] Koklu, Murat, and Ilker Ali Ozkan. "Multiclass classification of dry beans using computer vision and machine learning techniques." *Computers and Electronics in Agriculture* 174 (2020): 105507. UCI: 10.24432/C50S4B

[bc] Street, W. Nick, William H. Wolberg, and Olvi L. Mangasarian. "Nuclear feature extraction for breast tumor diagnosis." *Biomedical image processing and biomedical visualization.* Vol. 1905. SPIE, (1993). UCI: 10.24432/C5DW2B

[sp] Cranor, Lorrie Faith, and Brian A. LaMacchia. "Spam!." *Communications of the ACM* 41.8 (1998): 74-83. UCI: 10.24432/C53G6X

