# OpenReview forum: "Scientific Theory of a Black-Box: A Life Cycle-Scale XAI Framework Based on Constructive Empiricism"
_TMLR — Rejected by TMLR_

### Review · Reviewer_CQYP · 2026-03-17

**Summary Of Contributions:**

XAI methods (LIME, SHAP, Anchors, etc.) produce isolated, one-shot outputs. This paper proposes a living artefact, called a Scientific Theory of a Black Box (SToBB), that sits alongside a deployed model and grows as one queries it. Instead of running LIME ad hoc every time someone asks "why did the model predict X?", the SToBB maintains a surrogate model that is guaranteed to agree with the black-box on every observation, and keeps expanding this surrogate as new observations come in.

Philosophically, this is grounded in van Fraassen's Constructive Empiricism (CE): since we cannot know the "true reasoning" of the black-box, the framework only claims to reproduce all observed behaviour (empirical adequacy). This translates into three XAI requirements: (i) the surrogate must agree with the black-box on every recorded observation, (ii) when a new observation breaks adequacy, the surrogate must be revised via an explicit update, and (iii) all assumptions, construction choices, update history, and design rationale are documented for third-party scrutiny.

The framework specifies five components: an extensible observation base, a traceable hypothesis class, algorithmic components for construction/revision, an adequate surrogate, and documentation. Explanations for concrete stakeholder needs are obtained by querying the maintained record through interfaces (instead of isolated XAI method outputs). As a proof of concept, the authors instantiate a complete SToBB for a neural-network classifier on the Abalone dataset (tabular, 3-class) and introduce the Constructive Box Theoriser (CoBoT) algorithm, an online procedure that constructs and maintains an empirically adequate rule-based surrogate as observations accumulate.

**Audience:**

Yes

**Audience Explanation:**

**Strengths**

- **S1. Well-motivated structural gap.** XAI lacks a persistent, life-cycle-scale artefact for consolidating explanatory information. This is shown well: a literature survey of XAI methods combined with a review of governance frameworks (NIST, EU AI Act, ISO 42001) identifies that methods are applied ad hoc and produce isolated outputs.

- **S2. Principled philosophical framing.** The mapping from CE to XAI is carefully executed. Table 1 provides a clean alignment between CE concepts and SToBB design obligations.

- **S3. Separation of representation and presentation.** Decoupling the representational foundation from the interfaces as context-dependent presentation layers addresses the tension between fidelity and comprehensibility that plagues XAI. This is a design choice that deserves attention in its own right.

- **S4. Concrete algorithmic instantiation.** CoBoT maintains empirical adequacy incrementally as observations accumulate. The formal algorithm description (Algorithm 1) comes with guarantees: no overlap, unambiguous assignment, label consistency.

- **S5. Thoughtful documentation framework.** The question sheet in Appendix A provides a practical template that could be useful independently of the SToBB framing. This connects well to existing documentation practices (model cards, datasheets) and adds a new dimension by focusing on the explanatory artefact itself.

**Broader Impact Concerns:**

Given that it is primarily a conceptual/framework contribution with a proof-of-concept on public data (Abalone), no pressing ethical concerns arise from the work itself. However, in adopting a framework like this for governance, I could see adverse effects for privacy. The framework requires storing and accumulating input-output pairs over the model's life cycle. In domains with sensitive data (healthcare, finance, criminal justice), this creates a growing record that may conflict with data minimisation principles (e.g., GDPR Art. 5(1)(c)). The discussion mentions this in passing, but a Broader Impact section should flag it explicitly, especially given the paper's positioning around the EU AI Act.

**Claims And Evidence:**

No

**Claims Explanation:**

**Weaknesses**

- **W1. The CE apparatus may not be earning its keep.**

    - **W1.1.** The paper is essentially proposing a design pattern (maintain a versioned, empirically adequate surrogate with documentation) derived from philosophy of science. If the CE formalism is stripped away, what remains is: "keep a global surrogate that's always consistent with your observation log, update it incrementally, document everything" which sounds like good engineering practice.
    - **W1.2.** Currently the paper does not convincingly show that CE adds something beyond this (even though I think it may). Why is the philosophical apparatus required? A philosophical grounding should produce non-obvious design constraints, and the paper doesn't clearly identify any.
        - **W1.2.1.** For example: the claim that XAI should abandon the pretence of accessing "true model reasoning" and instead commit to a falsificationist-style process where adequacy is continually tested against observations. That's a substantive philosophical position with practical consequences; it reframes what fidelity _means_ for surrogates. This insight gets buried under the framework machinery.
        - **W1.2.2.** I think this is the paper's most interesting philosophical contribution (abandoning truth-claims about model internals in favour of falsificationist adequacy-testing), and it is never contrasted with the dominant implicit assumption in XAI, which is roughly a naïve realist one ("feature attributions tell us what the model is _really_ doing"). If the authors foregrounded that contrast, the CE apparatus would feel less ornamental. Currently, the paper spends more space on the mechanical CE-to-XAI translation than on articulating _why this epistemological shift matters in practice_.
    - **W1.3.** No comparison to other philosophy of science frameworks: scientific realism would demand more from the surrogate (approximate truth about internals); instrumentalism would demand less (just be useful, forget adequacy). Kuhn's paradigm structure, Lakatos's research programmes, or even Popperian falsificationism could each motivate a different version of SToBB with different obligations. The paper never argues why CE over these alternatives; it just picks CE and runs with it.

- **W2. The disagreement-problem claim is undersupported.**

    - **W2.1.** The claim is that SToBBs preempt the disagreement problem (apply LIME, SHAP, and Anchors to the same prediction and they can give conflicting explanations) because all interfaces query the same surrogate. While logically sound (same surrogate, so no source of disagreement), the paper never shows two interfaces producing outputs that would have disagreed under standard XAI methods but don't disagree under SToBB. That is the most natural experiment validating this claim, and it is absent.
    - **W2.2.** The claim is only valid within a single SToBB. If two teams build separate SToBBs for the same black-box (using different hypothesis classes or auxiliary measures), both can be empirically adequate but structurally incompatible, producing different explanations. The paper acknowledges Rashomon sets but doesn't connect this to the disagreement claim — which seems to push the disagreement problem to a higher level of abstraction (e.g., hypothesis class selection).
    - **W2.3.** Interfaces can still disagree at the presentation layer: a local interface might say "feature 2 was important," while the global interface might visually emphasise a different region of the input space. A user can perceive this as contradictory even though it isn't, strictly speaking, a logical inconsistency in the surrogate.

- **W3. Large gap between the framework's ambition and what is demonstrated.**

    - **W3.1.** The five-component architecture is claimed to be general across hypothesis classes and domains, but is only shown in one instance (axis-aligned boxes on 7-dimensional tabular data with a low-accuracy classifier).
    - **W3.2.** The framework claims life-cycle-scale utility, reuse across stakeholders, and extensibility — none of these are tested. No longitudinal deployment. No simulation of multi-phase usage. The proof of concept processes one batch of data in one phase.
    - **W3.3.** The claim that SToBBs complement existing documentation practices (model cards, datasheets) is supported only by references, with no concrete comparison or integration demonstration.
    - **W3.4.** The claim that SToBBs support AI governance and regulatory compliance references the EU AI Act, NIST RMF, and ISO 42001, but provides no mapping of specific regulatory requirements to SToBB components and no compliance case study.
    - **W3.5.** The claims are not wrong, but the evidence provided is insufficient.

- **W4. The proof-of-concept CoBoT has several issues.**

    - **W4.1.** The compression ratio (20 boxes / 4,177 samples) is reported but never evaluated against any baseline or alternative surrogate, e.g., existing rule learners (RIPPER, Anchors, decision lists). This makes the compression claim uninterpretable in isolation.
    - **W4.2.** CoBoT depends on LIME, which is stochastic: the same observation could be routed to different subspaces in different runs, placing it in a completely different boxsystem and potentially triggering different box structures downstream. The rejection criterion (discard observations where all LIME scores are negative) hints that the authors encountered degenerate LIME outputs, but they never report how often this happens or test what changes when you re-run with different seeds.
    - **W4.3.** Scalability: the paper claims CoBoT is "a generic procedure for tabular domains," but the algorithm's memory and compute costs are never analysed. With $d=7, k=3$ it's $\binom{7}{3} = 35$ subspaces, but this blows up combinatorially for larger $d$. The greedy merge procedure (which attempts to join compatible boxes after every update) presumably scales at least quadratically in the number of boxes per subspace. For life-cycle-scale use on real-world tabular data, this matters.
    - **W4.4.** The adaptive-$k$ mechanism (automatically trading off rule complexity vs. adequacy) was never triggered in the proof of concept ($k$ remained at 3 throughout). So this mechanism is proposed but untested.

- **W5. Interfaces are described but not evaluated.**

    - **W5.1.** Three example interfaces are provided (local, contrastive, global), but all are minor variations on box extraction. No user evaluation, no quantitative assessment of answer quality.
    - **W5.2.** The paper offers no formal constraints on interfaces beyond "must not directly modify the SToBB." What consistency guarantees hold across interfaces? Can an interface produce an explanation that contradicts the surrogate's structure (e.g., by adding background information that reverses the surrogate's conclusion)?
    - **W5.3.** The paper says interfaces "may answer questions regarding previously seen as well as unseen samples" and that unseen samples can trigger updates. This means the act of _querying_ the SToBB changes it. That is epistemically unusual; scientific theories don't mutate when you ask them a question. This deserves explicit discussion. (If I have misunderstood the mechanism, please clarify.)

- **W6. Consequences of strict empirical adequacy are underexplored.**

    - **W6.1.** For noisy or stochastic black-boxes (e.g., models with dropout at inference, or APIs with temperature > 0), strict binary adequacy becomes ill-defined. The paper assumes a fixed, deterministic black-box and could be quite fragile to violations of this assumption. This assumption should be made more prominent.
    - **W6.2.** A single adversarial or corrupted query can force a structural revision. There is no notion of robust adequacy or outlier handling. The paper's own observation-rejection criterion (LIME scores all < 0) is ad hoc and not grounded in the CE framework itself.
    - **W6.3.** Generalisation is explicitly dismissed: "the task is not to generalise beyond [the black-box] but to reproduce its behaviour on all available observations." This means the surrogate offers no epistemic value for new queries; there is no guarantee or even vocabulary for what happens when a user consults a SToBB about unseen inputs.
        - **W6.3.1.** The Abalone example works because the hypothesis class (axis-aligned boxes with LIME-selected subspaces) is expressive enough relative to a low-dimensional, low-accuracy classifier. For a high-accuracy model on complex data, achieving zero disagreement with a simple interpretable surrogate is questionable: either the surrogate becomes so complex it stops being interpretable, or $k$ keeps incrementing until the boxes are effectively singletons (no compression, no generalisation, no explanatory value). The paper should discuss this adequacy–interpretability tradeoff explicitly.
    - **W6.4.** Overall, the claim that empirical adequacy (binary, on all observations) is a meaningful and achievable requirement is not supported with sufficient evidence — only partially, in one instance.

- **W7. Missing related work.**

    - **W7.1.** Model reconciliation or explanation consolidation in the interpretability literature is not discussed.
    - **W7.2.** Work on global surrogate learning with fidelity constraints (e.g., Born-Again Trees [Breiman & Shang, 1996], TREPAN [Craven & Shavlik, 1996], or more recent distillation-based approaches) is absent.
    - **W7.3.** The Rashomon set literature beyond a brief mention of Marx et al. (2020) deserves deeper treatment. Given that multiple empirically adequate surrogates can coexist (as the paper acknowledges without fully exploring the implications), the relationship between SToBBs and Rashomon sets seems to run deeper than currently presented.
    - **W7.4.** Work on incremental/online rule learning (e.g., Very Fast Decision Rules, incremental RIPPER variants) is relevant to CoBoT's algorithmic contribution and should be discussed.

**Requested Changes:**

**Required for acceptance:**

1. **Justify the CE choice (addresses W1).** Either (a) explicitly argue why CE is preferred over competing philosophical frameworks (scientific realism, instrumentalism, Popperian falsificationism, Lakatos) for grounding SToBBs, identifying concrete design decisions that differ under each framework; or (b) foreground the epistemological shift — the abandonment of truth-claims about model internals in favour of falsificationist adequacy-testing — and contrast it with the implicit naïve realism in mainstream XAI. Currently, the most interesting philosophical contribution is buried. If the philosophical grounding does not demonstrably constrain the design beyond "good engineering," the paper should reduce its claims about what CE contributes.

2. **Narrow claims to match evidence (addresses W3, W6.4).** Life-cycle-scale utility, stakeholder reuse, extensibility, governance support, and preempting the disagreement problem are all claimed but not demonstrated. Either provide evidence (experiments, simulations, case studies) or reformulate these as motivated conjectures / future work. The paper should include an honest "Limitations" subsection that consolidates all caveats currently scattered across the text.

3. **Discuss the adequacy–interpretability tradeoff (addresses W6.3.1).** The paper should explicitly analyse under what conditions strict empirical adequacy is achievable with an interpretable surrogate. The current proof of concept (low-dimensional, low-accuracy classifier) does not stress-test this tradeoff. At minimum, a discussion of when and why the framework might fail is needed.

4. **Address stochasticity of LIME and its downstream effects on CoBoT (addresses W4.2).** Report the rejection rate, run sensitivity analysis across seeds, and discuss what the framework prescribes when the auxiliary measure itself is unreliable.


**Would strengthen the paper:**

5. **A second instantiation on a different domain and hypothesis class (addresses W3.1).** For example, a text classifier where the hypothesis class is a set of decision lists over token-level attributions, or an image classifier where the surrogate is prototype-based. This would enormously strengthen the generality claim. Currently, the reader has no evidence that the five-component architecture survives contact with anything beyond axis-aligned boxes on tabular data.

6. **Demonstrate meaningfully different interfaces (addresses W5.1).** Show that the same SToBB supports meaningfully different explanations for different stakeholders: e.g., a developer gets feature-importance rankings, an auditor gets coverage statistics over protected subgroups, a domain expert gets contrastive rules; all _provably consistent_ because they derive from the same adequate surrogate. Currently the three interfaces are all slight variations on "extract a box."

7. **Provide a disagreement-problem experiment (addresses W2.1).** Show two interfaces producing outputs that would have disagreed under standard XAI methods but don't disagree under SToBB. This is the most natural validation of the disagreement claim.

8. **Baseline comparisons for CoBoT (addresses W4.1).** Compare CoBoT's compression, fidelity, and rule quality against at least one established rule learner (RIPPER, Anchors, decision lists) to contextualise the reported numbers.

9. **Complexity analysis of CoBoT (addresses W4.3).** Provide time and space complexity of CoBoT's construction and update procedures as a function of $d$, $k$, number of observations, and number of boxes. Discuss scalability limits.

10. **Discuss the observation-on-query issue (addresses W5.3).** The fact that querying the SToBB can mutate it is epistemically unusual and practically consequential (e.g., ordering effects, audit reproducibility). This should be discussed explicitly, with reference to how it departs from the CE analogy where theories are not altered by the act of application.

11. **Expand related work (addresses W7).** Discuss global surrogates with fidelity constraints, incremental rule learning, and the deeper connection to Rashomon sets.

12. **Tighten the exposition (minor).** The paper is well written but diffuse in places: habitual hedging ("it is important to emphasise," "one can imagine," "it seems justified"), repeating "in this paper we," and the separation of representational foundation from presentation is explained in Sections 1, 4, 4.1, and 6. Section 3 (CE primer) could be compressed; philosophical details could move to the appendix. Consider leading with a concrete motivating scenario (e.g., a credit scoring model in production for 18 months where three different teams have run LIME, SHAP, and Anchors at different times, and an auditor arrives to find contradictory characterisations) before the philosophy (ML has a more applications-first culture than philosophy I assume where it's principles-first, depends on target audience). Notation: $B$, $b$, $\mathcal{B}$ are difficult to disambiguate at a glance.

---

> ### Author Response · Authors · 2026-04-02
> **Answer part 1**
>
> We thank the Reviewer for their thorough reading, detailed feedback, and valuable recommendations. Below, we address the comments raised by the Reviewer and indicate where they have been addressed in the revised manuscript. For the Reviewer's convenience, the corresponding changes are highlighted in blue and annotated in the margin with the relevant Reviewer ID (green).
>
> ### _Justify the CE choice (W1)_
> (**Addressed**) A new paragraph at the end of Section 2 now contrasts the choice of CE against other views in the philosophy of science and argues that the choice of the philosophical framework is not neutral: CE is preferred because it drops the unverifiable realist demand that the artefact approximates the model's true internal reasoning, and instead grounds all claims in observable, checkable behaviour, which we see as a prerequisite for a governance-oriented artefact that must be scrutinised by third parties. We believe this establishes that CE demonstrably constrains the design beyond good engineering.
> 	See the highlighted paragraph at the end of Section 2.
>
> ### _Narrow claims to match evidence (addresses W3, W6.4)_
> (**Done**) A dedicated Limitations subsection has been added to Section 6. We acknowledge that the single illustrative example does not fully demonstrate claims about lifecycle-scale utility, stakeholder reuse, and governance support in real deployment settings, and have reframed these as motivated conjectures pending more extensive instantiations.
> We maintain, however, that the example makes these claims plausible: reuse is demonstrated each time the surrogate covers a new observation (eg via probing by an interface) without an update, extensibility and stakeholder reuse follows from the framework's definition and the three interfaces operating over the same surrogate.
> See the highlighted Limitations subsection in Section 6.

---

> > ### Author Response · Authors · 2026-04-02
> > **Answer part 2**
> >
> > Continued:
> >
> > ### *Discuss the adequacy–interpretability tradeoff (addresses W6.3.1); discussion of when and why the framework might fail*
> > (**Addressed**) The revised manuscript now discusses the following concerns of the Reviewer: (1) when the hypothesis class is insufficiently expressive, the surrogate fragments toward singletons; (2) when auxiliary measures are systematically unreliable, the derived surrogate structure may be compromised; (3) when data retention constraints apply, the requirement to maintain all recorded observations creates tension with deletion.
> > For (1), we argue singletons are the correct honest outcome and signal where the hypothesis class needs reconsideration. For (2) and (3), dedicated paragraphs in the new Limitations subsection discuss what the framework prescribes and where open questions remain. A formal characterisation of when strict empirical adequacy is achievable with an interpretable surrogate is left for future work. See Sections 3.2 and 6.2.
> >
> > ### *Stochasticity of auxiliary measures*
> > (**Addressed**) The new Limitations paragraph on auxiliary measure reliability in Section 6.2 discusses what the CE framework prescribes: observations whose auxiliary measure is deemed unreliable can be excluded from the observation base, and the documentation obligation requires the choice of auxiliary measure and its quality criteria to be stated explicitly.
> > Regarding the specific auxiliary measure used in the illustration: CoBoT uses only the rank ordering of LIME attributions rather than their numerical values, which reduces sensitivity to small perturbations. A full sensitivity analysis across LIME seeds would be relevant for a thorough evaluation of CoBoT as an algorithm, which lies beyond the scope of the illustration. See the highlighted paragraph in Section 6.2.
> >
> > ### *Additional suggestions*
> > We have noted the remaining suggestions for strengthening the paper.
> > Regarding the observation-on-query issue, we argue in Sections 3.3 and 4.1 that querying on unseen inputs is a form of active testing coherent with the CE framework rather than an epistemically unusual property.
> > Regarding the disagreement problem experiment: the Reviewer acknowledges that disagreement does not persist when the surrogate is accepted as the authoritative reference, which is precisely the SToBB's premise. If two teams independently maintain separate SToBBs for the same black box, CE prescribes accepting one as the reference — operating two competing observation bases is not in the spirit of the framework. A disagreement-problem experiment would therefore demonstrate a property that follows from the framework's design by construction.
> > Regarding CoBoT-specific suggestions (related work on incremental rule learners, baseline comparisons, and complexity analysis) would be relevant for a thorough evaluation of CoBoT as an algorithmic contribution, which lies beyond the scope of the illustration; we have incorporated them as directions for future work in Section 6.3. A second instantiation on a different domain would substantially strengthen the generality claim and is the most natural next step for follow-up research.

---

### Review · Reviewer_GiiZ · 2026-03-18

**Summary Of Contributions:**

**Summary**

The paper introduces the "Scientific Theory of a Black-Box" (SToBB) framework, proposing a structural change in how Explainable AI (XAI) is applied. Relying on Constructive Empiricism, the authors propose shifting from isolated explanation methods toward maintaining a persistent and auditable explanatory artifact throughout a model's lifecycle. The framework enforces empirical adequacy, requiring the surrogate model to perfectly match all observed black-box behaviors. The authors provide a proof-of-concept via the Constructive Box Theoriser (CoBoT) algorithm, which builds and updates a rule-based surrogate for a neural network operating on tabular data.

**Strengths**

- The paper introduces a framework focused on AI governance and deployment. Instead of optimizing isolated explanations or approximating a model's unobservable reasoning, it prioritizes an auditable framework governed by continuous documentation and empirical consistency.

- Traditional XAI methods are frequently applied independently, which can lead to isolated and incompatible characterizations of the same model. The SToBB framework mitigates this by providing a single, shared representational foundation from which consistent explanations can be derived.

- The authors provide a practical implementation of the proposed philosophy. By introducing the CoBoT algorithm, they demonstrate how a SToBB can be instantiated, updated, and queried in practice to construct an empirically adequate rule-based surrogate.

**Weaknesses**

- The requirement that the surrogate must perfectly reproduce all available observations restricts feasible architectures. For more complex problems, it is reasonable to assume that this will also lead to a greater number of and more complex rules. Enforcing this could also lead, in the worst case scenario, to the need to retrain the surrogate from scratch for each new data point.

- The framework relies entirely on the premise that the black-box model is "fixed," meaning its decision behavior remains static throughout its life cycle. While the authors acknowledge this limitation in their discussion of related and future work, this passing treatment seems insufficient given the severity of the issue. I would have appreciated at least an idea of how to adapt the framework to handle evolving models.

- Based on how the CoBoT algorithm constructs its bounding boxes, it explicitly discards auxiliary measures (e.g., LIME scores) that fall below zero. This architectural choice limits the algorithm's utility by discarding valuable negative correlation data, information that standard interpretable models (such as decision trees or logistic regression) use highly effectively to map out exclusionary logic.

- The framework requires that all historical input-output data and auxiliary measures be saved in a continuously expanding "observation base". Storing all observational data over a model's operational life cycle poses significant storage and computational challenges for high-throughput systems, and directly conflicts with data minimization principles found in modern privacy regulations.

**Question**

Have you thought about using an ensemble or a hierarchy of surrogate models instead of a single one? A more complex model could be used to give explanations for the data points in which a simpler model fails, while keeping the explanations for the simpler samples more interpretable.

**Audience:**

Yes

**Audience Explanation:**

Yes. The findings of this paper will be of interest to a distinct subset of TMLR's audience, particularly researchers focused on XAI, AI governance, and model auditability.

The paper tackles the explainability problem from a novel and structured angle by grounding it in the philosophy of science (specifically, Constructive Empiricism). Instead of proposing another ad hoc feature attribution method, it formalizes a rigorous, life cycle-scale framework (SToBB) for maintaining and auditing trust in black-box models over time. This paradigm shift from single-shot explanations to continuous, empirically adequate documentation is relevant to current discussions surrounding AI safety and compliance.

**Broader Impact Concerns:**

No concerns.

**Claims And Evidence:**

Yes

**Claims Explanation:**

Yes, the central claims of the paper, specifically the conceptual need for a persistent, life cycle-scale XAI framework (SToBB), are well justified and represent a valuable contribution to the field. The authors successfully argue for this paradigm shift, and the provided proof-of-concept demonstrates that the theoretical framework can be operationalized.

While I remain skeptical of the practical efficacy and scalability of the Constructive Box Theoriser (CoBoT) algorithm, particularly given its restrictive reliance on exclusively positive auxiliary measures, this specific implementation is secondary to the paper's theoretical goals and does not diminish the value of the SToBB concept itself.

Furthermore, while the strict enforcement of empirical adequacy and the assumption of a fixed black-box model represent substantial limitations for real-world deployment, they do not invalidate the framework's foundational premise. These practical constraints are acceptable as topics for future research; however, given the severity of these operational bottlenecks, the paper would be strengthened by offering preliminary proposals on how future iterations of the framework might actually tackle evolving models and the inevitable explosion of surrogate complexity.

**Requested Changes:**

**Changes to the text**

The function f used on page 10 is defined only in the appendix. It should also be defined around the first time it appears.

**Changes to strengthen the work**

The authors could add a quick discussion on how to tackle changing models.

It would be interesting to see how the framework would scale with a more complex dataset.

---

> ### Author Response · Authors · 2026-04-02
>
> We thank the Reviewer for their thorough reading and valuable recommendations to further strengthen our work. Below, we address the comments raised by the Reviewer and indicate how and where they have been addressed in the revised manuscript. For the Reviewer's convenience, the corresponding changes are highlighted in blue and annotated in the margin with the relevant Reviewer ID (purple).
>
> ### *Have you thought about using an ensemble or a hierarchy of surrogate models instead of a single one? A more complex model could be used to give explanations for the data points in which a simpler model fails, while keeping the explanations for the simpler samples more interpretable.*
> (**Answer**) We take the motivation behind the question to concern the practical trade-off between complexity and comprehensibility for user-facing explanations. In the SToBB framework, that trade-off is handled primarily at the level of interfaces: the surrogate serves as the inspectable background representation, while interfaces derive stakeholder-appropriate explanations from it. This separation means that surrogate complexity and explanation comprehensibility are decoupled. A more complex surrogate does not necessarily produce less comprehensible explanations, as long as the interfaces are designed accordingly. We expect that the trade-off an interface makes can be made more transparent in the SToBB setting than in standard XAI methods. In LIME, for example, complexity reduction can be achieved through regularisation, but it is not clear what part of the black-box behaviour is "dropped". Using the CoBoT example, an interface could return only a subset of boxes while also stating (in some form) which observations are no longer represented by that abstraction.
>
> ### *Changes to text: define function f*
> (**Done**) Thank you for pointing that out, the function is now defined in place.
>
> ### *Discussion on changing models*
> (**Addressed**) We agree that this is a serious practical concern, and we have revised the paragraph in our discussion to sharpen the point.
> We also clarify why this dependence is principled: explanatory information such as predicted labels and auxiliary measures must be tied to a particular black-box instance. If such observations remained valid regardless of which model version produced them, they would not in fact describe that model’s behaviour. The revised paragraph therefore makes clear that model replacement requires a new SToBB, while still noting that previously collected input samples can still be reused as probes when rebuilding the observation base for the updated model.

---

### Review · Reviewer_aMTd · 2026-03-23

**Summary Of Contributions:**

The authors introduce the notion of a Scientific Theory of a Black-Box (SToBB), a persistent, auditable artefact that consolidates explanatory information about a fixed black-box model across its life cycle. Grounded in Constructive Empiricism (CE), a SToBB must satisfy empirical adequacy (the surrogate reproduces the black-box on all observations), adaptability (explicit update commitments), and auditability (documented assumptions and construction choices). Explanations for specific stakeholder needs are obtained by querying the maintained record through interfaces rather than producing isolated method outputs. As a proof-of-concept, the authors instantiate a SToBB for a neural-network classifier on the Abalone dataset using the Constructive Box Theoriser (CoBoT) algorithm, an online procedure that builds a rule-based surrogate from axis-aligned bounding boxes, using LIME attributions to determine relevant feature subspaces

**Additional Comments:**

In its current form the paper reads more as a position paper than an original research contribution for TMLR; the ratio of conceptual development to empirical validation is heavily skewed. I can see this paper either majorly revised as outlined above or orthogonally revised as a position paper.

**Audience:**

Yes

**Audience Explanation:**

The paper addresses a timely need, particularly as AI governance matures under the EU AI Act and related frameworks. The conceptual contribution, i.e. grounding XAI in Constructive Empiricism and distinguishing between explanatory information and explanation, is novel and could influence how the community thinks about surrogate-based explainability. However, the impact is currently limited by the gap between the ambition of the framework and the superficial nature of its empirical demonstration.

**Claims And Evidence:**

No

**Claims Explanation:**

The paper's claims are primarily conceptual, and while the CE-to-XAI translation is internally coherent (as far as I can tell, I am no philosophy expert), the proof-of-concept does not provide sufficient evidence to support the claimed practical benefits.The CoBoT instantiation is demonstrated on a single, small tabular dataset (Abalone, 4k samples, 7 features, 3 classes) with a neural network achieving only 60% accuracy.

 No comparison is made with any baseline, neither with standard global surrogate methods (e.g., a decision tree or rule list retrained periodically) nor with incremental/online interpretable learners. Given that the authors position the SToBB as superior to existing surrogate practices, a direct comparison demonstrating this advantage is essential. I am also not convinced by the treatment of the empirical adequacy requirement. The paper claims this is "stricter than typical surrogate modelling in XAI", but this claim is not fully supported. A sufficiently flexible hypothesis class trivially achieves perfect agreement; CoBoT's fallback to singleton boxes means adequacy is guaranteed by construction, which arguably weakens the value of the guarantee.
The compression diagnostic (Figure 1) hints at this tension but never formally analyzes it. Could the authors elaborate on what empirical adequacy actually buys the user beyond what a high-fidelity surrogate trained with standard methods provides? And under what conditions does the requirement become vacuous (e.g., when singletons dominate)?

Furthermore, the lifecycle and accumulation claims are not demonstrated. The proof-of-concept processes the dataset in a single pass. There is no simulation of a realistic multi-stakeholder, multi-stage scenario, no evidence that queries at different lifecycle stages benefit from previously accumulated information, and no evaluation of whether interface outputs derived from a SToBB actually exhibit greater consistency than independently applied XAI methods (the disagreement problem claim).

**Requested Changes:**

My main concerns are regarding missing baselines/related work, the depth of the empirical evaluation, and the under-examined implications of the empirical adequacy requirement.

Missing baselines and related work (critical).
A highly relevant body of work is missing from the evaluation. In terms of baselines, I am missing comparisons to standard global surrogate approaches to assess whether the SToBB/CoBoT actually provides tangible benefits. At minimum, the authors should compare against (a) a decision tree or rule list retrained on accumulated observations, and (b) an incremental rule learner (e.g., from the stream mining literature). The paper should also more carefully position itself relative to existing work on online/incremental interpretable models as the novelty of CoBoT relative to established incremental rule learners needs to be made explicit.

Empirical evaluation scope (critical).
The proof-of-concept on a single small dataset with a weak black-box (0.6 accuracy) is insufficient to support the paper's claims. I would like to see: (i) results on multiple datasets of varying complexity, including at least one moderately sized dataset, to demonstrate that the approach scales; (ii) quantitative evaluation of surrogate fidelity compared to baselines; and (iii) a concrete worked example showing how the SToBB is queried across different lifecycle stages and stakeholder needs, demonstrating the claimed accumulation and reuse benefits. On a related note, how do runtimes scale with input dimensionality and number of observations? The current setting  is too small to assess practical viability.

Analysis of empirical adequacy (critical).
The paper needs a more rigorous analysis of the expressiveness-adequacy trade-off. What is the relationship between the compression ratio and the epistemic value of the surrogate? Is there a formal sense in which requiring perfect fidelity on observed data provides guarantees beyond what a standard high-fidelity surrogate gives? The current treatment leaves the central theoretical contribution under-examined.

Interface evaluation (critical).
The three example interfaces are described algorithmically, but no user study, qualitative assessment, or even substantive worked examples with interpretation are provided. I would at minimum expect concrete, discussed examples of local and contrastive outputs that demonstrate interpretive value.

Dependence on LIME (important but not critical).
The observation space relies heavily on LIME attributions, which are known to be unstable. Since LIME attributions determine the subspace partitioning and hence the entire surrogate structure, instability in LIME could systematically undermine coherence. Could the authors elaborate on how sensitive the resulting SToBB is to the choice and stability of the auxiliary measure?

---

> ### Author Response · Authors · 2026-04-02
>
> We thank the Reviewer for their thorough reading and recognising that our works addresses a timely need in the XAI community. Below, we address the comments raised by the Reviewer and indicate how and where they have been addressed in the revised manuscript. For the Reviewer's convenience, the corresponding changes are highlighted in blue and annotated in the margin with the relevant Reviewer ID (orange).
>
> ### *Missing baselines and related work (critical)*, *Empirical evaluation scope (critical)*
> (**Addressed**) As noted in the general response above, the revision explicitly reframes CoBoT as an illustrative example rather than an independent algorithmic contribution.
> We further provide a reviewer-only appendix to address the specific concern that empirical adequacy via singletons is trivially achievable, comparing CoBoT against an unconstrained decision tree baseline. The results show that CoBoT maintains adequacy with near-zero singletons and compact structural growth, which the unconstrained baseline does not achieve. See Appendix C.
>
> ### *Analysis of empirical adequacy (critical), 1) Relationship between compression ratio and epistemic value of the surrogate? 2) Advantage empirical adequacy vs. high-fidelity surrogate?*
> (**Addressed**) 1) The compression ratio alone is an incomplete diagnostic, a high ratio achieved through genuine generalisation and one achieved through a small number of singletons are epistemically very different situations. The singleton count is the more informative measure, and the appendix shows that in the CoBoT illustration compression is achieved through generalisation rather than fragmentation. 2) The two types of guarantee are different in kind rather than one being stronger than the other. A high-fidelity surrogate minimises error on average but may be knowingly wrong on specific observed cases with no obligation to correct this. An empirically adequate surrogate makes a different promise: it is never knowingly wrong on any recorded observation, and any falsifying observation immediately triggers a revision. We argue that for a maintained lifecycle-scale artefact whose role is to serve as a reliable descriptive reference, this is the more relevant guarantee.
> We refer to the highlighted paragraphs in Section 3.2.
>
> ### *Interface evaluation (critical)*
> (**Addressed**) Rule-based explanations are an established format whose userfriendliness is well-documented in the XAI literature [1,2]. We added example outputs of the local and contrastive interfaces to the interface description (Appendix B) as they naturally fit with the documentation. The presentation and formatting of interface outputs, as well as any other information that needs to be included, is deliberately part of interface design rather than the SToBB framework itself, and will depend on the application context and target audience.
>
> ### *Dependence on LIME (important but not critical) and sensitivity to auxiliary measure*
> (**Addressed**) The new Limitations paragraph on auxiliary measure reliability in Section 6.2 discusses what the CE framework prescribes when an auxiliary measure is unreliable. Regarding the specific auxiliary measure used in the illustration: CoBoT uses only the rank ordering of LIME attributions rather than their numerical values, which reduces sensitivity to small numerical perturbations. A full sensitivity analysis across LIME seeds would be relevant for a thorough evaluation of CoBoT as an algorithm, which lies beyond the scope of the illustration. See the highlighted paragraph in Section 6.2.
>
>     [1] Ribeiro, Marco Tulio, Sameer Singh, and Carlos Guestrin. "Anchors: High-precision model-agnostic explanations." _Proceedings of the AAAI conference on artificial intelligence_. Vol. 32. No. 1. 2018.
>     [2] Lakkaraju, Himabindu, Stephen H. Bach, and Jure Leskovec. "Interpretable decision sets: A joint framework for description and prediction." _Proceedings of the 22nd ACM SIGKDD international conference on knowledge discovery and data mining_. 2016.

---

> > ### Comment · Reviewer_aMTd · 2026-04-07
> > **Remaining unaddressed critical issues**
> >
> > I thank the authors for their detailed response and for the effort invested in the revision. Thanks to the re-framing as a position paper, the paper reads more honestly about what it does and does not demonstrate.
> >
> > That said, three issues from my original review remain insufficiently addressed:
> >
> > **1. Empirical evaluation scope (critical, partially addressed)**
> >
> > My original request asked for results on multiple datasets of varying complexity, including at least one moderately sized dataset, to demonstrate that the approach scales. The revision retains a single small dataset (Abalone, 4,177 samples, 7 features) with a weak black-box (64% accuracy). The Limitations paragraph in Section 6.2 now honestly acknowledges that "claims about lifecycle-scale utility, stakeholder reuse, and extensibility … should be understood as motivated conjectures" , which I appreciate. However, even for a position paper, the illustrative example should inspire confidence that the framework can operate beyond a toy setting. A single low-dimensional tabular task with a near-chance-level classifier does not do this.
> >
> > Please add at least one additional dataset of meaningfully higher complexity (more features, more samples, or a stronger black-box) to show that CoBoT's structural growth, singleton count, and compression behaviour remain well-behaved as task difficulty increases. This need not be an exhaustive benchmark, just a second example with reasonable complexity that complements the Abalone illustration would be sufficient.
> >
> > **2. Missing incremental rule learner baseline (critical, not addressed)**
> >
> > My original review requested comparisons against two baselines: (a) a decision tree or rule list retrained on accumulated observations, and (b) an incremental rule learner from the stream-mining literature. The revision provides only (a), the decision tree comparison in Appendix C . The incremental rule learner is neither discussed nor compared against. This matters because CoBoT is an incremental rule learner by construction since it processes observations one at a time and updates its rule set online. The most natural baselines are therefore existing algorithms designed to do precisely this (e.g., VFDR, AMRules, or similar). Without such a comparison, the reader cannot assess whether CoBoT's compact structural growth and low singleton count are a property of the algorithm's design or simply a consequence of the easy task. I ask the authors to include at least one incremental/streaming rule learner as a second baseline.
> >
> > **3. Baseline comparison must be in the main paper (critical, not addressed)**
> >
> > The decision tree comparison is currently placed in a reviewer-only appendix with the explicit note "NOT PART OF THE PAPER" . This is problematic for two reasons. First, the claim that empirical adequacy is "stricter than typical surrogate modelling"  appears in the main text, but the only quantitative evidence supporting it is hidden from the reader. Second, the comparison directly addresses a core conceptual claim of the paper, namely that SToBB's adequacy requirement produces meaningfully different surrogate behaviour.  Please integrate the baseline comparison (both the decision tree and the  incremental rule learner) into the paper, ideally as part of or immediately following Section 5.

---

> ### Author Response · Authors · 2026-04-09
> **Addressing all raised issues**
>
> We thank the Reviewer for their continued engagement and giving more detail on their concerns.
> We provide an extended comparison across four datasets and two additional baselines in Appendix C, addressing the concerns about empirical scope, the incremental rule learner baseline, and the accessibility of quantitative evidence.
>
> ### *"1. Empirical Scope: Weakness of the black-box*
> (**Answer**)
> The black-boxes in the new settings achieve an accuracy above 90%.
> We want to note that the label distribution on Abalone is approximately uniform, for a 3 class problem 64% test-accuracy is considerably above near-chance-level. We further maintain the black-box accuracy is of secondary interest because the surrogate approximates the black-box output, not the ground truth labels.
>
> ### _show that CoBoT's structural growth, singleton count, and compression behaviour remain well-behaved as task difficulty increases_, _add a second example_, _2. Missing incremental rule learner baseline_
>
> (**Done**)
> We provide results for four datasets varying in dimensionality, sample size, and number of classes ("Beans", "Breastcancer", "Spambase", see Table in Appendix C), comparing CoBoT against a decision tree baseline and VFDR in two configurations ("default" and "permissive").
> AMRules, the other incremental rule learner referenced, is designed for regression and is not applicable to the classification setting here.
> _Beans (16 features, 13611 samples, 7 classes)_ directly addresses the request for a second example of meaningfully higher complexity: CoBoT achieves compact, stable surrogates with near-zero singletons, showing that structural growth remains well-behaved. The decision tree reaches a comparable success rate with higher structural complexity; both VFDR configurations fail to produce accurate rule coverage.
> _Breastcancer (30 features, 569 samples)_ represents a low-data regime: CoBoT produces around 20 boxes for 569 samples, requiring more cumulative updates than the decision tree, but the box count stabilises. At the end of the sequence all approaches reach comparable structural complexity, with CoBoT maintaining slightly fewer singletons. Both VFDR configurations again fail to achieve empirical adequacy.
> _Spambase (53 features, 4548 samples)_ illustrates the boundary condition discussed in Section 6.2: when observations are sparse relative to dimensionality, CoBoT fragments toward singletons. The decision tree also fails to converge, confirming the difficulty is a property of the task. Both VFDR configurations produce near-zero explicit rules with low success rates.
> Across all datasets and configurations, VFDR does not achieve empirical adequacy: its success rate never reaches 1.0, and the default rule absorbs --- and periodically re-absorbs after rule updates --- more observations than CoBoT produces singletons.
>
> The new results are provided in Appendix C.
>
>
> ### *3. Baseline comparison must be in the main paper*
> (**Answer**) We agree that the quantitative evidence should be accessible to the reader and not hidden in a reviewer-only document. We propose moving the comparison from the reviewer-only appendix to a published appendix, with a single reference sentence in Section 5 noting that a comparison against a decision tree baseline and VFDR is provided in Appendix C. This makes the evidence available to all readers without restructuring the main text around an algorithmic evaluation, which we fear would distract from the purpose of Section 5.

---

### Author Response · Authors · 2026-04-02
**Summary of changes**

We thank the reviewers for their careful and constructive reading of the manuscript. We are happy to read that the reviewers converge on a common observation: the conceptual transfer from Constructive Empiricism to XAI and the resulting SToBB framework are well grounded in its philosophical framing (CQYP) and the new direction of providing an explanatory base instead of single explanations (GiiZ) is of general interest for the TMLR audience (GiiZ, CQYP, aMTd). While the concrete, practical implementation of such a SToBB is generally appreciated (GiiZ, CQYP), we agree that the presentation of the CoBoT instantiation invited scrutiny beyond its intended scope as an illustration (CQYP, aMTd).
We have made adaptations to better stress that the role of CoBoT is to illustrate that the SToBB framework's five components can be instantiated and that empirical adequacy can be maintained incrementally.
The revision sharpens the presentation accordingly, foregrounding the framework and reframing CoBoT explicitly as an illustrative example.

In line with Reviewer aMTd's suggestion, the paper is revised as a position paper.

The revision
- strengthens the motivation for Constructive Empiricism in Section 2,
- expands the defence of empirical adequacy and its operational consequences in Section 3,
- restructures the final section into Discussion, Limitations, and Conclusion subsections with a dedicated Limitations paragraph,
- and adds a reviewer-only appendix with a baseline comparison.

Color coding:
- All changes are highlighted in blue and, if to specifically address a concern, annotated in the margin with the relevant reviewer ID.
- Strike-through text (in red) and other smaller changes indicate where the framing of the CoBoT contribution was corrected.

---

### Decision · Action_Editor_wZcq · 2026-04-29

**Recommendation:** Reject

**Audience:**

Yes

**Audience Explanation:**

There is a large audience for explainability methods and certified explanations / standards; however, the paper as it is does not meet the standards for evidence.

**Claims And Evidence:**

No

**Claims Explanation:**

The paper provides the following contributions:

- Proposing to enhance a trained ML model with an artifact (called a *scientific theory of a black box*) that provides the unique point of query for any stakeholder interested in explaining a prediction or auditing the model.
- Developing three properties that the artifact should guarantee in order for it to be useful.
- Grounding the entire discussion in the constructive empiricism (CE) philosophy of science (roughly: the artifact is to the model as a scientific theory is to the underlying phenomena it describes).
- Proposing one instantiation of the theory based on binning the input spaces into boxes.
- Testing the framework in a use case (Abalone dataset, small feedforward network).

We received three reviews which were quite negative. After rebuttal, one reviewer leans towards acceptance, one reviewer asks for modifications, one reviewer is still heavily concerned. I personally share the concerns of the last reviewer and I will use them to ground my decision.

- One reviewer argued that the entire paper is very close to "stick a global surrogate model on top of a trained model and certify it somewhat".
- One major difference is that the authors require "adequacy" for the surrogate model, which they define as "consistent with all available observations". This is a *huge* assumption and building an entire framework around it is very strange unless the authors can provide and validate an algorithm to achieve it.
- The algorithm they propose is relatively simple (boxing the input space of the classifier); it can blow up combinatorially in the number of features, ruling out its use in any advanced use case. In the main paper, the author validate it only on a hugely simplistic use case (Abalone) which cannot even be considered "toy".
- Upon the request of multiple reviewers they added some datasets and baselines, but they added them in a weird Appendix C labeled as "NOT PART OF THE PAPER (?) This section is not part of the paper and is provided for the reviewers’ reference only."
- The reviewers also lamented that many claims of the paper (e.g., "lifecycle-scale utility and stakeholder reuse without qualification") are not validated.

Overall, the idea of connecting an explanation to a scientific theory of the model is interesting, and the discussion flows nicely. However, unless the authors can provide a reasonable global surrogate model and validate it empirically (multiple datasets, large-scale experiments, ablations, baselines), everything else feels "empty" and discursive.

**Resubmission Of Major Revision:**

The authors may consider submitting a major revision at a later time.